# Rethinking and Reweighting the Univariate Losses for Multi-Label Ranking: Consistency and Generalization

**Guoqiang Wu**[*], **Chongxuan Li**[*], **Kun Xu**, **Jun Zhu**[†]

Dept. of Comp. Sci. & Tech., Institute for AI, State Key Lab for Intell. Tech. & Sys.,
Tsinghua-Bosch Joint Center for ML, BNRist Center, Tsinghua University, Beijing, 100084 China
{guoqiangwu90, chongxuanli1991, kunxu.thu}@gmail.com,
dcszj@tsinghua.edu.cn

## Abstract

The (partial) ranking loss is a commonly used evaluation measure for multi-label classification, which is usually optimized with convex surrogates for computational efficiency. Prior theoretical efforts on multi-label ranking mainly focus on (Fisher) consistency analyses. However, there is a gap between existing theory and practice — some inconsistent pairwise losses can lead to promising performance, while some consistent univariate losses usually have no clear superiority in practice. To take a step towards filling up this gap, this paper presents a systematic study from two complementary perspectives of consistency and generalization error bounds of learning algorithms. We theoretically find two key factors of the distribution (or dataset) that affect the learning guarantees of algorithms: the instance-wise class imbalance and the label size $c$. Specifically, in an extremely imbalanced case, the algorithm with the consistent univariate loss has an error bound of $O(c)$, while the one with the inconsistent pairwise loss depends on $O(\sqrt{c})$ as shown in prior work. This may shed light on the superior performance of pairwise methods in practice, where real datasets are usually highly imbalanced. Moreover, we present an inconsistent reweighted univariate loss-based algorithm that enjoys an error bound of $O(\sqrt{c})$ for promising performance as well as the computational efficiency of univariate losses. Finally, experimental results confirm our theoretical findings.

## 1 Introduction

Multi-Label Classification (MLC) [1] is an important task, in which each instance is associated with multiple labels simultaneously. It has been widely applied to text categorization [2], bioinformatics [3], multimedia annotation [4], information retrieval [5]. Various measures [6, 7] have been developed to evaluate MLC's performance from diverse aspects owing to its complexity. Among them, the *ranking loss* (RL) (or *partial ranking loss*, PRL) [2, 8] is a commonly used measure in practice (or in theory). Formally, the RL calculates the fraction of pairs that a positive label does not precede a negative label according to the rank given by a *score function* (or predictor). Minimizing such a loss is usually referred to as Multi-Label Ranking (MLR) [9], which is the consideration in this paper.

Since RL is non-convex and discontinuous, existing methods [6] seek to optimize certain convex surrogate losses for computational efficiency. These surrogate losses can be divided into two main categories: pairwise ones [8] and univariate ones [9], which have their own advantages and limitations in terms of computational costs, theory and empirical performance.

---

[*]Equal contribution. G. Wu is now at School of Software, Shandong University and C. Li is now at Gaoling School of AI, Renmin University of China. The work was done when they were at Tsinghua University.
[†]Corresponding author

35th Conference on Neural Information Processing Systems (NeurIPS 2021).

Table 1: Summary of the main theoretical results. Contributions of this paper are highlighted in red.

| Algorithm | Surrogate loss | Generalization bound | | Consistency[a] | Computation |
|---|---|---|---|---|---|
| | | extremely imbalanced | balanced | | |
| $\mathcal{A}^{pa}$ | pairwise ($L_{pa}$) | $\hat{R}^{pa}_S(f) + O(\sqrt{\frac{c}{n}})$ | $\hat{R}^{pa}_S(f) + O(\sqrt{\frac{1}{n}})$ | $\times$ | $O(c^2)$ |
| $\mathcal{A}^{u_1}$ | univariate ($L_{u_1}$) | $c\hat{R}^{u_1}_S(f) + O(\sqrt{\frac{c^2}{n}})$ | $2\hat{R}^{u_1}_S(f) + O(\sqrt{\frac{1}{n}})$ | $\times$ | $O(c)$ |
| $\mathcal{A}^{u_2}$ | univariate ($L_{u_2}$) | $(c-1)\hat{R}^{u_2}_S(f) + O(\sqrt{\frac{c^2}{n}})$ | $\frac{c}{2}\hat{R}^{u_2}_S(f) + O(\sqrt{\frac{1}{n}})$ | $\checkmark^b$ | $O(c)$ |
| $\mathcal{A}^{u_3}$ | reweighted univariate ($L_{u_3}$) | $\hat{R}^{u_3}_S(f) + O(\sqrt{\frac{c}{n}})$ | $\hat{R}^{u_3}_S(f) + O(\sqrt{\frac{1}{n}})$ | $\times$ | $O(c)$ |
| $\mathcal{A}^{u_4}$ | reweighted univariate ($L_{u_4}$) | $\hat{R}^{u_4}_S(f) + O(\sqrt{\frac{c^2}{n}})$ | $\hat{R}^{u_4}_S(f) + O(\sqrt{\frac{1}{n}})$ | $\times$ | $O(c)$ |

[a]This is in terms of the PRL. Besides, these surrogate losses are all inconsistent w.r.t. the RL.
[b]This is for the cases where the base loss is the exponential, logistic, least squared or squared hinge loss.

Computationally, the pairwise losses, defined over pairs of positive and negative labels, have a complexity depending on $O(c^2)$ ($c$ is the number of labels), while the univariate ones enjoy a complexity depending on $O(c)$. The superiority of the latter is significant when $c$ is large. Theoretically, the pairwise losses are not (Fisher) consistent w.r.t. the RL or the PRL [8], while, remarkably, certain univariate ones are consistent w.r.t. the PRL [9, 8]. Empirically, however, the consistent univariate losses usually have no significant superiority compared to the inconsistent pairwise losses [9]. In fact, we observed that the former under-perform the latter on 10 MLR benchmarks (see results in Table 3). Towards filling the gap between the existing theory and practice, this paper attempts to rigorously answer the following questions: *Why inconsistent pairwise losses usually achieve better performance than consistent univariate losses in practice? How to improve the univariate loss, which is preferable due to its computational efficiency?* A natural explanation of this gap is that although the (Fisher) consistency [10, 11] provides valuable insights in the asymptotic cases, it cannot fully characterize the behaviour of a surrogate loss when the number of training samples is not sufficiently large and the hypothesis space is not realizable.

Therefore, this paper presents a systematic study in a complementary perspective of generalization error bounds [12] besides the consistency. We theoretically find two key factors of the distribution (or dataset) that affect the learning guarantees of algorithms: the instance-wise class imbalance and the label number $c$. Given extremely imbalanced data (i.e., the worst case), we prove that the consistent univariate losses based algorithms lead to an error bound of $O(c)$ while the pairwise losses based ones enjoy an error bound depending on $O(\sqrt{c})$ [13], which explains the empirical behaviour better on real datasets which are usually highly imbalanced (see Table 3). Further, we present two *reweighted surrogate univariate losses* that employ carefully designed penalties for positive and negative labels. Then, we analyze their consistency and generalization bounds of the corresponding algorithms. Surprisingly, though not consistent, one of them enjoys an error bound depending on $O(\sqrt{c})$ in the worst case, which is nearly the same as the pairwise losses, and retains the computational efficiency. For balanced data (i.e., the best case), we also find that all these surrogate univariate losses share the same learning guarantees, with no dependence on $c$ which is the same as the surrogate pairwise ones. Notably, in this case, these bounds are different from the classical probably approximately correct (PAC) ones [14, 12, 15] which hold for all distributions (i.e. the worst case). Our main theoretical results are summarized in Table 1. Experimental results also confirm our theoretical findings.

## 2 Preliminaries

In this section, we first introduce the problem setting of MLC and MLR. Then, we present the evaluation measures, risk, and regret of MLR.

**Notations**. Let boldface lower and capital case letters denote vectors (e.g., $\mathbf{a}$) and matrices (e.g., $\mathbf{A}$) respectively. For a matrix $\mathbf{A}$, $\mathbf{a}_i$, $\mathbf{a}^j$ and $a_{ij}$ denote its $i$-th row, $j$-th column, and $(i,j)$-th element respectively. For a vector $\mathbf{a}$, $a_i$ denote its $i$-th element. For a set, $|\cdot|$ denotes the cardinality. $[\![\pi]\!]$ denotes the indicator function, i.e., $[\![\pi]\!] = 1$ if the proposition $\pi$ holds and 0 otherwise.

## 2.1 Problem Setting

Let $\mathbf{x} \in \mathcal{X} \subset \mathbb{R}^d$ and $\mathbf{y} \in \mathcal{Y} \subset \{-1, +1\}^c$ denote the input and output respectively, where $d$ is the feature dimension, $c$ is the label size. $y_j = 1$ (or $-1$) indicates that the associated $j$-th label is relevant (or irrelevant). Given a training set $S = \{(\mathbf{x}_i, \mathbf{y}_i)\}_{i=1}^n$ of $n$ samples i.i.d. drawn from a distribution $P$ over $\mathcal{X} \times \mathcal{Y}$, the original goal of MLC is to learn a multi-label classifier $H : \mathbb{R}^d \to \{-1, +1\}^c$.

To solve MLC, a common approach is to first learn a vector-based *score function* (or predictor) $f = [f_1, ..., f_c] : \mathbb{R}^d \to \mathbb{R}^c$ and then get the classifier $H$ by a thresholding function. Multi-Label Ranking (MLR) aims to learn the best predictor from the finite training data in terms of some ranking-based measures, which is our consideration in this paper.

## 2.2 Evaluation Measures

To evaluate the performance of different approaches for MLR, many measures have been developed. Here we focus on two widely-used measures in practice (or theory), which are defined below.[3]

**Ranking Loss (RL):**

$$L_r^{0/1}(f(\mathbf{x}), \mathbf{y}) = \frac{1}{|S_{\mathbf{y}}^+||S_{\mathbf{y}}^-|} \sum_{(p,q) \in S_{\mathbf{y}}^+ \times S_{\mathbf{y}}^-} [\![f_p(\mathbf{x}) \le f_q(\mathbf{x})]\!], \tag{1}$$

where $S_{\mathbf{y}}^+$ (or $S_{\mathbf{y}}^-$) denotes the relevant (or irrelevant) label index set induced by $\mathbf{y}$.

**Partial Ranking Loss (PRL):**[4]

$$L_{pr}^{0/1}(f(\mathbf{x}), \mathbf{y}) = \frac{1}{|S_{\mathbf{y}}^+||S_{\mathbf{y}}^-|} \sum_{(p,q) \in S_{\mathbf{y}}^+ \times S_{\mathbf{y}}^-} \left[ [\![f_p(\mathbf{x}) < f_q(\mathbf{x})]\!] + \frac{1}{2} [\![f_p(\mathbf{x}) = f_q(\mathbf{x})]\!] \right]. \tag{2}$$

Note that the only difference between these two measures is the penalty when $f_p(\mathbf{x}) = f_q(\mathbf{x})$ holds. Besides, it is easy to verify that RL upper bounds PRL, i.e. $L_{pr}^{0/1}(f(\mathbf{x}), \mathbf{y}) \le L_r^{0/1}(f(\mathbf{x}), \mathbf{y})$. Although these two measures are almost the same in practice for the evaluation of one algorithm, they have different consistency properties for some surrogate losses theoretically [8].

## 2.3 Risk and Regret

Since RL is non-convex and discontinuous, often leading to NP-hard problems [16], it is optimized with convex surrogates in practice for computational efficiency. Define a surrogate loss $L_\phi : \mathbb{R}^c \times \{-1, +1\}^c \to \mathbb{R}_+$, where $\phi$ indicates the specific surrogate loss and will be detailed in the next section. Besides, define a vector-based predictor class $\mathcal{F} = \{f : \mathcal{X} \to \mathbb{R}^c\}$. For a predictor $f \in \mathcal{F}$, its true $(0/1)$ expected risk, surrogate expected risk, and surrogate empirical risk are defined as:

$$R_{0/1}(f) = \mathop{\mathbb{E}}_{(\mathbf{x}, \mathbf{y}) \sim P}[L^{0/1}(f(\mathbf{x}), \mathbf{y})], \; R_\phi(f) = \mathop{\mathbb{E}}_{(\mathbf{x}, \mathbf{y}) \sim P}[L_\phi(f(\mathbf{x}), \mathbf{y})], \; \hat{R}_S(f) = \frac{1}{n} \sum_{i=1}^n L_\phi(f(\mathbf{x}_i), \mathbf{y}_i).$$

Besides, we use a superscript (i.e., $pr$ or $r$) to distinguish the risks for specific measures. Further, for convenience, we denote the expected risk conditioned on an instance $\mathbf{x}$ (i.e., the conditional risk) as:

$$R(f|\mathbf{x}) = \mathop{\mathbb{E}}_{\mathbf{y} \sim P(\mathbf{y}|\mathbf{x})}[L(f(\mathbf{x}), \mathbf{y})|\mathbf{x}] = \sum_{\mathbf{y}} L(f(\mathbf{x}), \mathbf{y})P(\mathbf{y}|\mathbf{x}), \tag{3}$$

where $L$ denotes the true or surrogate loss. Thus, the expected risk of $f$ is $R(f) = \mathbb{E}_{\mathbf{x} \sim P(\mathbf{x})}[R(f|\mathbf{x})]$.

For each $\mathbf{x}$, given the conditional distribution $P(\mathbf{y}|\mathbf{x})$, we can get its optimal predictions as follows:[5]

$$f^*(\mathbf{x}) = \arg\min_{\mathbf{a} \in \mathbb{R}^c} \sum_{\mathbf{y}} L(\mathbf{a}, \mathbf{y})P(\mathbf{y}|\mathbf{x}), \tag{4}$$

where $f^*$ is called the *Bayes predictor* w.r.t. the loss $L$. Besides, the expected risk of $f^*$ (i.e., $R(f^*)$) is called the *Bayes risk*, which is the minimal expected risk w.r.t. the loss $L$ and denoted by $R^*$ for

---

[3]Our definition is over one sample and can be averaged over multiple samples.

[4]Minimizing the PRL is equivalent to maximize the instance-AUC.

[5]Notably, the optimal predictions can be not just one value but a set with many elements that share the same minimal conditional risk.

convenience. Then, we can define the regret (a.k.a. excess risk) of a predictor $f$ w.r.t. the true and surrogate loss as follows.

$$Reg_{0/1}(f) = R_{0/1}(f) - R^*_{0/1}, \qquad Reg_\phi(f) = R_\phi(f) - R^*_\phi. \tag{5}$$

Besides, we also use a superscript (i.e., $pr$ or $r$) to distinguish the regrets for specific measures. Moreover, let $\hat{f}_n$ denote the learned predictor from finite training data $S$. Note that our goal is to find a predictor $\hat{f}_n$ that achieves the minimal true regret (i.e. $Reg_{0/1}(\hat{f}_n)$) as possible as it can.

## 3 Methods

In this section, we first introduce several specific surrogate losses. Then, we present their associated learning algorithms.

### 3.1 Surrogate Losses

To optimize the RL, it is natural to employ the convex surrogate pairwise loss [2, 3, 17, 18] as:

$$L_{pa}(f(\mathbf{x}), \mathbf{y}) = \frac{1}{|S_\mathbf{y}^+||S_\mathbf{y}^-|} \sum_{(p,q) \in S_\mathbf{y}^+ \times S_\mathbf{y}^-} \ell(f_p(\mathbf{x}) - f_q(\mathbf{x})). \tag{6}$$

where the base (margin-based) convex loss $\ell(z)$ can be defined in various popular forms, such as the exponential loss $\ell(z) = e^{-z}$, the logistic loss $\ell(z) = \ln(1 + e^{-z})$, the hinge loss $\ell(z) = \max\{0, 1-z\}$, and squared hinge loss $\ell(z) = (\max\{0, 1-z\})^2$. A common property is that the base convex surrogate loss upper bounds the original $0/1$ loss,[6] i.e., $[\![z \leq 0]\!] \leq \ell(z)$.

Besides, the surrogate univariate loss, which primarily aims to optimize Hamming loss [19, 13], can also be viewed as a surrogate loss for the RL, which is defined as follows:

$$L_{u_1}(f(\mathbf{x}), \mathbf{y}) = \frac{1}{c} \sum_{j=1}^c \ell(y_j f_j(\mathbf{x})). \tag{7}$$

Notably, $L_{u_1}$ cannot strictly upper bound the RL, i.e. $L_r^{0/1}(f(\mathbf{x}), \mathbf{y}) \not\leq L_{u_1}(f(\mathbf{x}), \mathbf{y})$. Previous work presents the consistent surrogate univariate loss [9, 8] w.r.t. PRL, which is defined as follows:

$$L_{u_2}(f(\mathbf{x}), \mathbf{y}) = \frac{1}{|S_\mathbf{y}^+||S_\mathbf{y}^-|} \sum_{j=1}^c \ell(y_j f_j(\mathbf{x})). \tag{8}$$

Again, the consistent surrogate loss $L_{u_2}$ cannot strictly upper bound the RL either. Notably, when the surrogate loss strictly upper bounds the $0/1$ loss, the true $(0/1)$ risk can be upper bounded by the surrogate risk too, which is crucial for its generalization analysis. Thus, we present two *reweighted convex surrogate univariate losses*, which strictly upper bound RL and PRL, defined as below.

$$L_{u_3}(f(\mathbf{x}), \mathbf{y}) = \frac{\sum_{p \in S_\mathbf{y}^+} \ell(y_p f_p(\mathbf{x}))}{|S_\mathbf{y}^+|} + \frac{\sum_{q \in S_\mathbf{y}^-} \ell(y_q f_q(\mathbf{x}))}{|S_\mathbf{y}^-|}, \tag{9}$$

$$L_{u_4}(f(\mathbf{x}), \mathbf{y}) = \frac{1}{\min\{|S_\mathbf{y}^+|, |S_\mathbf{y}^-|\}} \sum_{j=1}^c \ell(y_j f_j(\mathbf{x})). \tag{10}$$

For a clear presentation, we will formally discuss their relationships in the next section.

### 3.2 Learning Algorithms

In the following, we consider the kernel-based learning algorithms which have been widely used in practice [3, 19, 20, 21, 22] and in theory [13] for MLC. Besides, our following analyses can be extended to other forms of hypothesis class, such as neural networks [23]. Let $\mathbb{H}$ be a reproducing kernel Hilbert space (RKHS) induced by the kernel function $\kappa$, where $\kappa : \mathcal{X} \times \mathcal{X} \to \mathbb{R}$ is a Positive Definite Symmetric (PDS) kernel. Let $\Phi : \mathcal{X} \to \mathbb{H}$ be a feature mapping associated with $\kappa$. The kernel-based hypothesis class can be defined as follows.

$$\mathcal{F} = \left\{ \mathbf{x} \mapsto \mathbf{W}^\top \Phi(\mathbf{x}) : \mathbf{W} = (\mathbf{w}_1, \dots, \mathbf{w}_c)^\top, \|\mathbf{W}\| \leq \Lambda \right\}, \tag{11}$$

---

[6]The original logistic loss can be easily changed to $\ell(z) = \log_2(1 + 2^{-z})$ to satisfy this condition.

where $\|\mathbf{W}\|$ denotes $\|\mathbf{W}\|_{\mathbb{H},2} = (\sum_{j=1}^c \|\mathbf{w}_j\|_{\mathbb{H}}^2)^{1/2}$ for convenience.

Here we consider the following five learning algorithms with the corresponding aforementioned surrogate losses. They can be formulated as follows with $\lambda$ denoting a trade-off hyper-parameter:

$$\mathcal{A}^{pa} : \min_{\mathbf{W}} \frac{1}{n} \sum_{i=1}^n L_{pa}(f(\mathbf{x}_i), \mathbf{y}_i) + \lambda \|\mathbf{W}\|^2, \tag{12}$$

$$\mathcal{A}^{u_k} : \min_{\mathbf{W}} \frac{1}{n} \sum_{i=1}^n L_{u_k}(f(\mathbf{x}_i), \mathbf{y}_i) + \lambda \|\mathbf{W}\|^2, k = 1, 2, 3, 4. \tag{13}$$

## 4 Theoretical Analyses

In this section, we present generalization error bounds of the learning algorithms presented before and consistency analyses of the corresponding surrogate losses.

Consistency analyses aim to answer the question of whether the $(0/1)$ expected risk of the learned function converges to the Bayes risk [11, 8] when samples goes to infinity. It can provide valuable insights for learning from infinite (or relatively large) data with an unconstrained hypothesis class. In comparison, generalization bounds may offer more insights for learning from finite data with a constrained hypothesis class. It is possible to analyze the finite sample case in the perspective of consistency if a regret bound holds. However, we argue that it typically results in a looser bound w.r.t. $c$ (i.e. $O(c\sqrt{c})$) compared to the generalization bound as detailed in Appendix A.

### 4.1 Generalization Analyses

Technically, we mainly utilize the Rademacher complexity [24] and the vector-contraction inequality [25], following the recent work [13]. Note that, advanced techniques [26], such as local Rademacher complexity [27] can be utilized to get tighter bound w.r.t. $n$ (i.e. $O(\frac{1}{n})$) by modifying the algorithm. But this is not our focus and we aim to analyze learning guarantees of these algorithms in the same framework and compare them fairly. Besides, we find that their learning guarantees are much dependent on the distribution imbalance. Thus, we first give the following definition.

**Definition 1** (Instance-wise class balanced and extremely imbalanced distribution). *For a distribution $P$ for MLC, it is said to be instance-wise class balanced if for any $(\mathbf{x}, \mathbf{y})$ sampled from $P$, $|S_{\mathbf{y}}^+| = |S_{\mathbf{y}}^-|$ holds; it is said to be instance-wise class extremely imbalanced if for any $(\mathbf{x}, \mathbf{y})$ sampled from $P$, $|S_{\mathbf{y}}^+| = 1$ or $|S_{\mathbf{y}}^-| = 1$ holds.*[7]

Then, we introduce the common mild assumptions for the subsequent analyses.

**Assumption 1** (The common assumptions).

*(1) The hypothesis class is defined in Eq.(11).*
*(2) The training dataset $S = \{(\mathbf{x}_i, \mathbf{y}_i)\}_{i=1}^n$ is sampled i.i.d. from the distribution $P$, where $\exists\, r > 0$, it satisfies $\kappa(\mathbf{x}, \mathbf{x}) \leq r^2$ for all $\mathbf{x} \in \mathcal{X}$.*
*(3) The base (convex) loss $\ell(z)$ is $\rho$-Lipschitz continuous and bounded by $B$.* [8]

Then we provide the properties (including the Lipschitz constants) of surrogate losses (See Appendix B.1). The Lipschitz constants of surrogates characterize the relationship between the Rademacher complexities [24] of the loss class and the hypothesis class based on the vector-contraction inequality [25], which plays a central role in the generalization analysis. Next, we analyze the relationship between true and surrogate losses as follows, which to prove learning guarantees of algorithms.

**Lemma 1** (The relationship between true and surrogate losses; full proof in Appendix B.2). *For the RL and its surrogate losses, the following inequalities hold for any $f \in \mathcal{F}$ and any sample $(\mathbf{x}, \mathbf{y})$:*

$$L_r^{0/1}(f(\mathbf{x}), \mathbf{y}) \leq L_{pa}(f(\mathbf{x}), \mathbf{y}) \leq L_{u_3}(f(\mathbf{x}), \mathbf{y}) \leq L_{u_4}(f(\mathbf{x}), \mathbf{y}) \leq (c-1)L_{u_2}(f(\mathbf{x}), \mathbf{y}), \tag{14}$$

$$\min\{|S_{\mathbf{y}}^+|, |S_{\mathbf{y}}^-|\}L_{u_2}(f(\mathbf{x}), \mathbf{y}) \leq L_{u_3}(f(\mathbf{x}), \mathbf{y}) \leq \max\{|S_{\mathbf{y}}^+|, |S_{\mathbf{y}}^-|\}L_{u_2}(f(\mathbf{x}), \mathbf{y}). \tag{15}$$

---

[7]In this paper we call them balanced or extremely imbalanced distribution (or dataset) for short.

[8]Note that, the widely-used hinge and logistic loss are both 1-Lipschitz continuous. Although the exponential, and squared hinge losses are not globally Lipschitz continuous, they are locally Lipschitz continuous.

*Besides, note that $L_{u_2}$ cannot strictly upper bound $L_r^{0/1}$ and $L_{pa}$, i.e.,*

$$L_r^{0/1}(f(\mathbf{x}), \mathbf{y}) \not\leq L_{u_2}(f(\mathbf{x}), \mathbf{y}), \ L_{pa}(f(\mathbf{x}), \mathbf{y}) \not\leq L_{u_2}(f(\mathbf{x}), \mathbf{y}).$$

From this lemma, we can observe that when an algorithm minimizes $L_{u_2}$, it also optimizes an upper bound of $L_r^{0/1}$ (or $L_{pa}$) which depends on $O(c)$. Besides, $L_{u_3}$ and $L_{u_4}$ strictly upper bound $L_r^{0/1}$ (or $L_{pa}$). These upper bounds of $L_r^{0/1}$ (or $L_{pa}$) would help to give learning guarantees of corresponding learning algorithms w.r.t. the (partial) ranking loss in the subsequent analyses. Furthermore, we can get the relationship between true and surrogate expected risks as follows.

**Lemma 2** (The relationship between true and surrogate expected risks; full proof in Appendix B.3). *For any $f \in \mathcal{F}$ and any distribution $P$, the following inequalities hold:*

$$R_{0/1}^{pr}(f) \leq R_{0/1}^r(f) \leq R_{pa}(f) \leq R_{u_3}(f) \leq R_{u_4}(f) \leq (c-1)R_{u_2}(f), \tag{16}$$

$$R_{0/1}^r(f) \not\leq R_{u_2}(f), \ R_{pa}(f) \not\leq R_{u_2}(f). \tag{17}$$

**Remark 1.** *From this lemma, we can see that, among the surrogate expected risks, the pairwise surrogate expected risk $R_{pa}$ provides the tightest upper bound of the true expected risk for the same hypothesis. Thus, to study the learning guarantees of algorithms w.r.t. $L_r^{0/1}$ (or $L_{pr}^{0/1}$), we can first analyze their counterparts w.r.t. $L_{pa}$.* [9]

From above analyses, we find the instance-wise class imbalance affects the Lipschitz constants of the surrogates and the relationship between these surrogates. Besides, it is hard to fully characterize the imbalance in real data. Thus, we consider two extremely cases w.r.t. the imbalance in the following.

### 4.1.1 The extremely imbalanced distribution (worst case)

In this section, we analyze the learning guarantees of these algorithms for the extremely imbalanced distribution. In this case, the Lipschitz constants (See Lemma B.1 in Appendix B.1) of the surrogates are largest. Therefore, these error bounds can be viewed as the worst cases and thus hold for all the distributions just like the classical probably approximately correct (PAC) bounds [14, 12, 15].

First, we analyze the learning guarantee of $\mathcal{A}^{u_2}$, as follows.

**Theorem 1** (Learning guarantee of $\mathcal{A}^{u_2}$ for extremely imbalanced distribution (worst case)). *Assume the loss $L_\phi = (c-1)L_{u_2}$, where $L_{u_2}$ is defined in Eq.(8). Besides, Assumption 1 holds and suppose $P$ is extremely imbalanced. Then, for any $\delta > 0$, with probability at least $1 - \delta$ over S, the following generalization bound holds for all $f \in \mathcal{F}$:*

$$R_{0/1}^r(f) \leq R_{pa}(f) \leq (c-1)R_{u_2}(f) \leq (c-1)\hat{R}_S^{u_2}(f) + 2\sqrt{2}\rho c\sqrt{\frac{\Lambda^2 r^2}{n}} + 3Bc\sqrt{\frac{\log\frac{2}{\delta}}{2n}}. \tag{18}$$

The full proof is in Appendix B.3.1. From this theorem, we can see that $\mathcal{A}^{u_2}$ has a learning guarantee w.r.t. $L_{pa}$ (or $L_r^{0/1}$) which depends on $O(c)$.

Then, we provide the learning guarantee of $\mathcal{A}^{u_3}$ in the following theorem.

**Theorem 2** (Learning guarantee of $\mathcal{A}^{u_3}$ for extremely imbalanced distribution (worst case)). *Assume the loss $L_\phi = L_{u_3}$, where $L_{u_3}$ is defined in Eq.(9). Besides, Assumption 1 holds and suppose $P$ is extremely imbalanced. Then, for any $\delta > 0$, with probability at least $1 - \delta$ over S, the following generalization bound holds for all $f \in \mathcal{F}$:*

$$R_{0/1}^r(f) \leq R_{pa}(f) \leq R_{u_3}(f) \leq \hat{R}_S^{u_3}(f) + 4\sqrt{2}\rho\sqrt{\frac{c\Lambda^2 r^2}{n}} + 6B\sqrt{\frac{\log\frac{2}{\delta}}{2n}}. \tag{19}$$

The full proof is in Appendix B.3.2. From this theorem, remarkably, we can see that $\mathcal{A}^{u_3}$ has a learning guarantee w.r.t. $L_{pa}$ (or $L_r^{0/1}$) which depends on $O(\sqrt{c})$, which enjoys the same order as the algorithm $\mathcal{A}^{pa}$ [13]. Moreover, we find that the learning guarantee of $\mathcal{A}^{u_4}$ w.r.t. $L_{pa}$ (or $L_r^{0/1}$) depends on $O(c)$ (See Appendix B.3.3).

---

[9]Note that, instead of directly bounding the 0/1 risk, we can treat our results as the upper bounds of the pairwise risk, allowing it greater than 1.

#### 4.1.2 The balanced distribution (best case)

Here we analyze the learning guarantees of algorithms for balanced distributions. In this case, the Lipschitz constants (See Lemma B.1 in Appendix B.1) of the surrogates are smallest. Thus, these error bounds can be viewed as the best case over all distributions, which is different from the classical PAC bounds that hold for all distributions. First, we can find that all algorithms with the univariate losses are exactly the same, which should share the same learning guarantee, and it is confirmed by the following theorem.

**Theorem 3** (Learning guarantee of $\mathcal{A}^{u_k}$, $k = 1, 2, 3, 4$ for balanced distribution (best case); full proof in Appendix B.4.1). *Assume the loss $L_\phi = 2L_{u_1} = \frac{c}{2}L_{u_2} = L_{u_3} = L_{u_4}$, where they are defined in Section 3.1. Besides,* Assumption 1 *holds and suppose $P$ is balanced. Then, for any $\delta > 0$, with probability at least $1 - \delta$ over $S$, the following generalization bound holds for all $f \in \mathcal{F}$:*

$$R_{0/1}^r(f) \le R_{pa}(f) \le R_{u_3}(f) = \frac{c}{2}R_{u_2}(f) \le \frac{c}{2}\hat{R}_S^{u_2}(f) + 4\sqrt{2}\rho\sqrt{\frac{\Lambda^2 r^2}{n}} + 6B\sqrt{\frac{\log\frac{2}{\delta}}{2n}}, \quad (20)$$

*where $2\hat{R}_S^{u_1}(f) = \frac{c}{2}\hat{R}_S^{u_2}(f) = \hat{R}_S^{u_3}(f) = \hat{R}_S^{u_4}(f)$.*

From Theorem 3, we can observe that the model complexity term in this bound has no dependence on $c$. Notably, the same learning guarantee for these learning algorithms also confirms the validity of our analyses and the tightness of these bounds. Moreover, $\mathcal{A}^{pa}$ also has an error bound independent of $c$ for balanced distribution (see Appendix B.4.2).

#### 4.1.3 Comparison

For generalization analyses, a tighter upper bound usually suggests probably better performance [12][10]. In this paper, all algorithms are analyzed in the same framework and we also consider the extremely imbalanced case where the upper bounds of the Lipschitz constants in different surrogates are tight. Given that the MLC distributions (or datasets) are usually highly imbalanced, it is relatively safe to evaluate the performance of the algorithms in theory by comparing their upper bounds.

We now compare the algorithms considering the cases with (approximately) imbalanced distributions.

- $\mathcal{A}^{pa}$ **vs** $\mathcal{A}^{u_2}$. $\mathcal{A}^{pa}$ usually has a tighter bound than $\mathcal{A}^{u_2}$. In particular, given the same hypothesis space, it is usually easier to train $\hat{R}_S^{pa}$ than other univariate losses, making $\hat{R}_S^{pa}$ smaller than others including $\hat{R}_S^{u_2}$.[11] Besides, for the model complexity terms (i.e. the last two terms), $\mathcal{A}^{pa}$ has an error bound of $O(\sqrt{c})$ while $\mathcal{A}^{u_2}$ depends on $O(c)$.
- $\mathcal{A}^{u_3}$ **vs** $\mathcal{A}^{u_2}$. Similarly, we argue that $\mathcal{A}^{u_3}$ usually has a tighter bound than $\mathcal{A}^{u_2}$. For the first risk term, $(c-1)\hat{R}_S^{u_2}$ is usually comparable or even larger than $\hat{R}_S^{u_3}$.[12] For the model complexity term, $\mathcal{A}^{u_3}$ has an error bound of $O(\sqrt{c})$ while $\mathcal{A}^{u_2}$ depends on $O(c)$.

Overall, the tighter bound of $\mathcal{A}^{pa}$ (and $\mathcal{A}^{u_3}$) than $\mathcal{A}^{u_2}$ indicates that $\mathcal{A}^{pa}$ (and $\mathcal{A}^{u_3}$) would probably perform better than $\mathcal{A}^{u_2}$, especially for a large label space (See Fig. 3 in Appendix E). Experimental results on imbalanced benchmark datasets in Table 3 confirm our analyses.

In contrast, given balanced distributions, $\mathcal{A}^{u_2}$ and $\mathcal{A}^{u_3}$ share the same learning guarantee. Thus, in approximately balanced cases, $\mathcal{A}^{u_2}$ perform probably similarly to $\mathcal{A}^{u_3}$.

Note that our aforementioned formal generalization analyses are mainly for two extreme cases about the distribution imbalance. As for the in-between cases of the imbalance, it is indeed highly nontrivial to consider a continuous changing imbalance level of the distribution when instances may have different numbers of positive labels, and we leave it as an important future direction.

Nevertheless, our framework can be applied to the cases where each instance has the same number of positive labels, denoted as $c_p$ (See Appendix B.5 for details). Here we set $c_{min} = \min\{c_p, c - c_p\}$ for the clarity of following discussions. $\frac{c_{min}}{c}$ directly reflects the imbalance level of the distribution.

---

[10]When comparing bounds, it is usually more reasonable to compare the order of dependent variables rather than comparing the absolute values.

[11]Although we can not formally express the claim, we empirically observed it in experiments.

[12]In some cases, the first risk term may be bigger than 1 but we can still take insights from the error bound through the dependent variables of the model complexity.

Note that the extremely imbalanced case ($c_p = 1$ or $c_p = c - 1$) and the balanced one ($c_p = c/2$) are included. According to these results, we can observe that $\mathcal{A}^{pa}$ has an error bound of $O(\sqrt{\frac{c}{c_{min}}})$ that is the same as $\mathcal{A}^{u_3}$, while $\mathcal{A}^{u_2}$ depends on $O(\frac{c}{c_{min}})$. Therefore, in an imbalanced case (not necessarily the extremely imbalanced ones), $\mathcal{A}^{pa}$ and $\mathcal{A}^{u_2}$ usually have better bounds than $\mathcal{A}^{u_3}$, which probably indicates their better performance over $\mathcal{A}^{u_3}$ in practice.

Besides, when $c$ is in an extreme scale, such as millions or more, the $O(c)$ computational time complexity for loss functions is prohibitive, and practitioners often resort to various negative sampling methods. While this is not the focus of this paper, our theoretical results indeed indicate that the univariate loss $L_{u_3}$ would probably enjoy better performance than other univariate ones w.r.t. (partial) ranking loss, thus its estimator by use of negative sampling methods may be preferred in practice.

## 4.2 Consistency Analyses

Following [9, 8], we consider the *general partial ranking loss* as follows:

$$L_{gpr}^{0/1}(f(\mathbf{x}), \mathbf{y}) = \alpha_{\mathbf{y}} \sum_{(p,q) \in S_{\mathbf{y}}^+ \times S_{\mathbf{y}}^-} \left[ [\![ f_p(\mathbf{x}) < f_q(\mathbf{x}) ]\!] + \frac{1}{2} [\![ f_p(\mathbf{x}) = f_q(\mathbf{x}) ]\!] \right], \qquad (21)$$

where $\alpha_{\mathbf{y}}$ is a positive penalty. Note that the losses in Eq. (2) is a special case of Eq. (21) with $\alpha_{\mathbf{y}} = \frac{1}{|S_{\mathbf{y}}^+||S_{\mathbf{y}}^-|}$ respectively. The general ranking loss can be defined in a similar way as Eq. (21). For clarity and generality, we define the *general reweighted univariate surrogate loss* as follows:

$$L_u(f(\mathbf{x}), \mathbf{y}) = \sum_{j=1}^c ([\![ y_j = +1 ]\!] \beta_{\mathbf{y}}^+ + [\![ y_j = -1 ]\!] \beta_{\mathbf{y}}^-) \ell(y_j f_j(\mathbf{x})), \qquad (22)$$

where $\beta_{\mathbf{y}}^+$ and $\beta_{\mathbf{y}}^-$ are penalties for the positive and negative labels respectively. We assume $\beta_{\mathbf{y}}^+ \beta_{\mathbf{y}}^- > 0$ for convenience in our analyses. Note that the penalties can be different and all univariate surrogate losses presented in Section 3.1 are special cases of Eq. (22). (See Table 2 for details.)

Let $\mathcal{B}_L(\mathbf{x}, P(\mathbf{y}|\mathbf{x}))$ denote the set of the Bayes predictors of a loss $L$ given a sample $\mathbf{x}$ and a conditional distribution $P(\mathbf{y}|\mathbf{x})$. Remarkably, [8] provided a sufficient and necessary condition for determining whether a surrogate loss to be (Fisher) consistent w.r.t. the (partial) ranking loss or not (See Lemma 3 in Appendix C). Checking the consistency of a new surrogate loss using Lemma 3 takes additional efforts, because one has to enumerate all conditional distributions. For the loss in Eq. (22), we present more intuitive characterization that only involves the penalties in Theorem 4, considering various base losses (See Proposition 1 in Appendix C for the results of the Hinge loss):

**Theorem 4** (Necessary condition for the consistency of Eq. (22) w.r.t. Eq. (21) with exponential, logistic or squared hinge loss)**.** *A general reweighted univariate surrogate loss in Eq. (22) with* $\ell(z) = e^{-z}$, $\ell(z) = \ln(1 + e^{-z})$ *or* $\ell(z) = (\max\{0, 1-z\})^2$ *is consistent w.r.t. the general partial ranking loss in Eq. (21) only if* $\exists \tau > 0$, $\beta_{\mathbf{y}}^+ \beta_{\mathbf{y}}^- = \tau \alpha_{\mathbf{y}}^2$ *for all* $\mathbf{y}$ *such that* $1 - c \leq \sum_{1 \leq j \leq c} y_j \leq c - 1$.

For clarity, see full proof in Appendix C. When $c \leq 3$, the penalties of $L_{u_1}$, $L_{u_3}$ and $L_{u_4}$ may coincide with that of $L_{u_2}$ up to a multiplicative constant. When $c \geq 4$, it is straightforward to construct counter examples that violate the necessary condition in Theorem 4 and obtain the Corollary 1 as follows.

**Corollary 1** (Inconsistency of $L_{u_1}$, $L_{u_3}$ and $L_{u_4}$ w.r.t. Eq. (2) with exponential, logistic or squared hinge loss)**.** *If* $c \geq 4$, $L_{u_1}$, $L_{u_3}$ *and* $L_{u_4}$ *with* $\ell(z) = e^{-z}$ *or* $\ell(z) = \ln(1 + e^{-z})$ *or* $\ell(z) = (\max\{0, 1-z\})^2$ *are inconsistent w.r.t. the partial ranking loss in Eq. (2).*

We further show the inconsistency of the general reweighted univariate loss in Eq. (22) w.r.t. the general partial ranking loss in Eq. (21) with hinge loss. Note that this includes the inconsistency of $L_{u_1}$, $L_{u_3}$ and $L_{u_4}$ w.r.t. Eq. (2).

An immediate conclusion from Corollary 1 is that $L_{u_1}$, $L_{u_3}$ and $L_{u_4}$ are inconsistent w.r.t. the RL in Eq. (1) because $\mathcal{B}_{L_r^{0/1}}(\mathbf{x}, P(\mathbf{y}|\mathbf{x})) \subset \mathcal{B}_{L_{pr}^{0/1}}(\mathbf{x}, P(\mathbf{y}|\mathbf{x}))$ [8].

Table 2: The penalties of the specific univariate losses w.r.t. the general reweighted form in Eq. (22).

| Loss | $L_{u_1}$ | $L_{u_2}$ | $L_{u_3}$ | $L_{u_4}$ |
|------|-----------|-----------|-----------|-----------|
| $1/\beta_{\mathbf{y}}^+$ | $c$ | $|S_{\mathbf{y}}^+||S_{\mathbf{y}}^-|$ | $|S_{\mathbf{y}}^+|$ | $\min\{|S_{\mathbf{y}}^+|, |S_{\mathbf{y}}^-|\}$ |
| $1/\beta_{\mathbf{y}}^-$ | $c$ | $|S_{\mathbf{y}}^+||S_{\mathbf{y}}^-|$ | $|S_{\mathbf{y}}^-|$ | $\min\{|S_{\mathbf{y}}^+|, |S_{\mathbf{y}}^-|\}$ |

Compared to existing work [9, 8], although Theorem 4 is negative, it considers surrogate losses in a more general reweighted form, i.e. Eq. (22), which may be of independent interest.

## 5 Related Work

Here we mainly review the theoretical work relevant to this paper in MLC and MLR.

**Consistency.** [8] studied the consistency of various surrogate losses w.r.t. Hamming and (partial) ranking loss. Remarkably, [9] presented an explicit regret bound w.r.t. partial ranking loss for $L_{u_2}$. Extensive work investigated the consistency w.r.t. other measures, especially the F-measure. For instance, [28] provided justifications and connections w.r.t. the F-measure using the empirical utility maximization (EUM) framework and the decision-theoretic approach (DTA) in binary classification, which were applied to the optimization of the macro-F measure in MLC. Further, [29] studied connections and differences between these two frameworks and clarified the notions of consistency w.r.t. many complex measures (e.g., the Jaccard measure) in binary classification.[13] Besides, prior work [30, 31] studied the consistency of the F-measure in MLC from the DTA perspective via different approaches to estimate the conditional distribution $P(\mathbf{y}|\mathbf{x})$. [32] devoted to the study of consistent multi-label classifiers w.r.t. various measures under the EUM framework. [33] investigated the multi-label consistency of various reduction methods w.r.t. precision@$k$ and recall@$k$ measures.

**Generalization analysis.** [13] studied the generalization bounds of the algorithms based on $L_{pa}$ and $L_{u_1}$ w.r.t. the ranking loss besides the Hamming loss and Subset Accuracy.

Technically, our generalization and consistency analyses for MLC mainly follow the prior work [13] and [8], respectively.

We mention that a specific form of Eq. (9) with base hinge loss has been used as a part of prior work [34], which achieves excellent empirical results in MLC. In comparison, this paper considers a more general form of such reweighted surrogate losses and provides formal consistency and generalization analyses, which have not been investigated in the literature to our knowledge.

## 6 Experiments

As a theoretical work, the primary goal of experiments is to corroborate our theoretical findings rather than showing the empirical superiority of the proposed algorithm. Therefore, we evaluate all algorithms in Section 3 w.r.t. the ranking loss on 10 widely-used benchmark datasets with various domains and sizes of label and data. The detailed statistics and access of these datasets are given in Appendix D. For all algorithms, we utilize linear models with the base logistic loss for simplicity and fair comparison. Besides, we use the same efficient stochastic algorithm (i.e. SVRG-BB [35]) to solve these convex optimization problems. Moreover, we search the hyper-parameter $\lambda$ for all algorithms from a wide range of $\{10^{-8}, 10^{-7}, \cdots, 10^2\}$ using 3-fold cross validation on each dataset.[14]

The experimental results are summarized in Table 3 (see Appendix E for complete results with standard deviations). First, we observe that $\mathcal{A}^{pa}$ and $\mathcal{A}^{u_3}$ outperform the others especially $\mathcal{A}^{u_2}$ on almost all benchmarks. It verifies our generalization analyses: in extremely imbalanced distributions case, $\mathcal{A}^{pa}$ and $\mathcal{A}^{u_3}$ usually have tighter bound than others, especially $\mathcal{A}^{u_2}$, where benchmarks are usually highly imbalanced as shown in Appendix E (Fig. 2). Besides, such results do not contradict with the consistency results since two assumptions in consistency analyses are violated in the real settings. The first one is that the Bayes predictor may not be linear and the second one is that the number of samples may not be sufficient to achieve the Bayes predictor, which explains the relatively weaker results of $\mathcal{A}^{u_2}$.

To further study the effect of $c$, we evaluate $\mathcal{A}^{u_2}$ and $\mathcal{A}^{u_3}$ on highly imbalanced semi-synthetic datasets with randomly selected $c$ based on the delicious dataset (See Fig. 3 in Appendix E for details).

---

[13]Note that, our generalization and consistency analyses are both under the EUM framework.
[14]Our code is available at `https://github.com/GuoqiangWoodrowWu/MLR-theory`.

Table 3: Ranking loss of all five algorithms on benchmark datasets. On each dataset, the top two algorithms are highlighted in bold and the top one is labelled with $^\dagger$. The number in the parentheses denotes the label size $c$ of the corresponding dataset. Besides, "-" means that $\mathcal{A}^{pa}$ takes more than a week using a 48-core CPU server on the delicious dataset (see Fig. 1 in Appendix E for quantitative results).

| Dataset | $\mathcal{A}^{pa}$ | $\mathcal{A}^{u_1}$ | $\mathcal{A}^{u_2}$ | $\mathcal{A}^{u_3}$ | $\mathcal{A}^{u_4}$ |
|---|---|---|---|---|---|
| emotions(6) | $\mathbf{0.1511}^\dagger$ | 0.1538 | 0.1587 | **0.1530** | 0.1616 |
| image(5) | $\mathbf{0.1625}^\dagger$ | 0.1642 | 0.1653 | **0.1645** | 0.1678 |
| scene(6) | $\mathbf{0.0696}^\dagger$ | 0.0809 | 0.0821 | **0.0768** | 0.0806 |
| yeast(14) | $\mathbf{0.1766}^\dagger$ | 0.1768 | 0.1785 | **0.1767** | 0.1816 |
| enron(53) | $\mathbf{0.0682}^\dagger$ | 0.0724 | **0.0696** | 0.0698 | 0.0715 |
| rcv1-subset1(101) | $\mathbf{0.0361}^\dagger$ | 0.0418 | 0.0392 | **0.0368** | 0.0391 |
| bibtex(159) | **0.0516** | 0.0545 | 0.0551 | $\mathbf{0.0401}^\dagger$ | 0.0538 |
| corel5k(374) | **0.1081** | 0.1091 | 0.1099 | $\mathbf{0.1063}^\dagger$ | 0.1096 |
| mediamill(101) | **0.0395** | 0.0402 | 0.0412 | $\mathbf{0.0389}^\dagger$ | 0.0405 |
| delicious(983) | - | **0.0960** | 0.0974 | $\mathbf{0.0946}^\dagger$ | 0.0978 |
| Time complexity | $O(c^2)$ | $O(c)$ | $O(c)$ | $O(c)$ | $O(c)$ |

We find that $\mathcal{A}^{u_3}$ would probably perform better than $\mathcal{A}^{u_2}$ with larger $c$, which also confirms our theoretical findings: in the extremely imbalanced case, $\mathcal{A}^{u_3}$ has tighter bound than $\mathcal{A}^{u_2}$ w.r.t. $c$. [15]

Furthermore, to study whether the upper bound for the generalization error can reflect on the true generalization error reasonably well, we conduct experiments on the semi-synthetic delicious datasets, where the result is shown in Appendix E (Table 3). We find that despite the absolute values of probabilistic upper bounds (PUB) might not reflect on the true generalization error reasonably well, the PUB (and expected surrogate pairwise risk) can still offer valuable insights into these learning algorithms under the same analysis framework (See Appendix E for details).

## 7 Conclusion and Discussion

This paper presents a systematic study from two complementary perspectives of consistency and generalization bounds of algorithms in multi-label ranking. Theoretically, we find that the instance-wise class imbalance and the label size of the datasets play an important role in their learning guarantees. In particular, for the extremely imbalanced case, existing algorithms with consistent univariate losses have an error bound of $O(c)$ while the ones with inconsistent pairwise losses depend on $O(\sqrt{c})$ [13], which can explain the superior performance of pairwise methods on highly imbalanced datasets in practice. Moreover, we present one inconsistent reweighted surrogate univariate loss-based algorithm which enjoys an error bound $O(c)$ which is nearly the same as the pairwise ones. For the balanced case, all the algorithms with the univariate losses share the same error bound with no dependence on $c$, which is the same as the pairwise ones. Finally, empirical results corroborate our theoretical findings.

For generalization analyses, we consider two extreme cases about the instance-wise class imbalance of the data distribution, and it may be better for fine-grained analyses through integrating other properties of the distribution. Besides, how to make these bounds tighter will inspire new learning algorithms. For the consistency analyses, we only consider classical Fisher consistency, and it may be better to take the hypothesis set into account, such as H-consistency [36].

## Acknowledgements

This work was supported by NSFC Projects (Nos. 61620106010, 62061136001, 61621136008, U1811461, U19B2034, U19A2081), Beijing NSF Project (No. JQ19016), Tsinghua-Bosch Joint Center for Machine Learning, Beijing Academy of Artificial Intelligence (BAAI), a grant from Tsinghua Institute for Guo Qiang, Tiangong Institute for Intelligent Computing, and the NVIDIA NVAIL Program with GPU/DGX Acceleration.

---

[15]Notably, it is nontrivial to directly plot the complexity rate of $c$ on semi-synthetic data. This is because when we change $c$, we also change the data distribution, while the generalization errors among different data distributions are not directly comparable.

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
