# Appendix for "Rethinking and Reweighting the Univariate Losses for Multi-Label Ranking: Consistency and Generalization"

## Contents

35th Conference on Neural Information Processing Systems (NeurIPS 2021).

**Notations**. For a square matrix, $\text{Tr}(\cdot)$ denotes the trace operator. For a function $g : \mathbb{R} \to \mathbb{R}$ and a matrix $\mathbf{A} \in \mathbb{R}^{m \times n}$, define $g(\mathbf{A}) : \mathbb{R}^{m \times n} \to \mathbb{R}^{m \times n}$, where $g(\mathbf{A})_{ij} = g(a_{ij})$.

## A Two perspectives of generalization bounds and consistency analyses

Firstly, we want to highlight the complementary roles of the two perspectives. Recall that our goal is to find a predictor $\hat{f}_n$ learned from finite training data that achieves the minimal true regret $Reg_{0/1}(\hat{f}_n)$. In the following, we will decompose the regret appropriately for clear discussions.

For generalization analyses, the true regret can be decomposed into the following terms w.r.t. the $0/1$ loss.

$$Reg_{0/1}(\hat{f}_n) = R_{0/1}(\hat{f}_n) - R^*_{0/1} = \underbrace{\left[ R_{0/1}(\hat{f}_n) - \inf_{g \in \mathcal{F}} R_{0/1}(g) \right]}_{\text{estimation error}} + \underbrace{\left[ \inf_{g \in \mathcal{F}} R_{0/1}(g) - R^*_{0/1} \right]}_{\text{approximation error}}, \quad (1)$$

where $\mathcal{F}$ is the constrained function class that real learning algorithms utilize. For a given distribution $P(\mathbf{x}, \mathbf{y})$ and a specific measure, $R^*_{0/1}$ is fixed. Besides, $\inf_{g \in \mathcal{F}} R_{0/1}(g)$ depends on the size of $\mathcal{F}$ and is fixed for a given $\mathcal{F}$. Thus, in this case, the original goal becomes to minimize $R_{0/1}(\hat{f}_n)$ as possible as it can. In Section 4.1, we present the generalization error bounds of the learning algorithms to provide learning guarantees for $R_{0/1}(\hat{f}_n)$ through bounding the surrogate risk $R_\phi(\hat{f}_n)$.[1] However, these error bounds cannot exactly tell the size of the gap between $R_{0/1}(\hat{f}_n)$ and $R_\phi(\hat{f}_n)$.

Consistency analyses aim to answer the question whether the $(0/1)$ expected risk of the learned function converges to the Bayes risk [1, 2], i.e., when $n \to \infty$, $R_\phi(\hat{f}_n) \to R^*_\phi \implies R_{0/1}(\hat{f}_n) \to R^*_{0/1}$. If a loss is consistent, a regret bound [1, 3] as follows is preferable. Namely, for all measurable function $f$ (including $\hat{f}_n$) and valid joint distribution $P(\mathbf{x}, \mathbf{y})$, the following holds:

$$R_{0/1}(f) - R^*_{0/1} \leq \psi^{-1}(R_\phi(f) - R^*_\phi), \quad (2)$$

where $\psi$ is an invertible function such that for any sequence $(\theta_i)$ in $[0, 1]$, $\psi(\theta_i) \to 0$ if and only if $\theta_i \to 0$ [1]. Prior work [3] shows that $\psi^{-1}(\theta) = O(c\sqrt{c})\sqrt{\theta}$ with logistic and exponential loss in MLR. Besides, when learning in the real setting (with finite data), the surrogate regret of $\hat{f}_n$ can be decomposed into the following two terms w.r.t. the surrogate loss.

$$R_\phi(\hat{f}_n) - R^*_\phi = \underbrace{\left[ R_\phi(\hat{f}_n) - \inf_{g \in \mathcal{F}} R_\phi(g) \right]}_{\text{estimation error}} + \underbrace{\left[ \inf_{g \in \mathcal{F}} R_\phi(g) - R^*_\phi \right]}_{\text{approximation error}}, \quad (3)$$

where the estimation error is due to finite data size, and the approximation error is due to the choice of $\mathcal{F}$. Notably, the consistency analysis [1] neglects these two errors since it allows $P(\mathbf{y}|\mathbf{x})$ known in the infinite data setting and assumes that the hypothesis class $\mathcal{F}$ is over all measurable functions.

In summary, consistency can provide valuable insights for learning from infinite data (or data of relatively large $n$ w.r.t. $c$) with an unconstrained hypothesis class, while generalization bounds can offer more insights for learning from finite data with a constrained hypothesis class.

Note that, there is another way that consistency can also explain the finite-sample effect for consistent losses, which results in a 0/1 excess risk bound (for $\mathcal{A}^{u_2}$) that depends on $O(c\sqrt{c})$ and nearly $O(n^{-\frac{1}{4}})$. This is obtained by combining the regret bound in Eq.(2) with $\psi^{-1}(\theta) = O(c\sqrt{c})\sqrt{\theta}$ [3] and the same analysis framework as performed in Theorems 1 and 2 to upper bound the estimation error in Eq.(3). In contrast, by combination of Eq.(1) and Theorem 1, this paper obtains a 0/1 excess risk bound (for $\mathcal{A}^{u_2}$) which depends on $O(c)$ and nearly $O(n^{-\frac{1}{2}})$. Therefore, in terms of $c$ and $n$, our result (for $\mathcal{A}^{u_2}$) is tighter than that obtained by the aforementioned way and the main claim of the paper remains.

---

[1]Notably, this requires that the surrogate loss $L_\phi$ strictly upper bounds the $0/1$ loss $L_{0/1}$ to make $R_{0/1} \leq R_\phi$.

# B Generalization Analyses

## B.1 Proof of Lemma B.1

**Lemma B.1** (The properties of surrogate losses). *Assume that the base (convex) loss $\ell(z)$ is $\rho$-Lipschitz continuous and bounded by $B$. Then, the following holds for the extremely imbalanced distribution.*

*(1) $L_{pa}(f(\mathbf{x}), \mathbf{y})$ in Eq. (6) is $\rho$-Lipschitz w.r.t. the first argument and bounded by $B$.*

*(2) $L_{u_2}(f(\mathbf{x}), \mathbf{y})$ in Eq. (8) is $\frac{\rho\sqrt{c}}{c-1}$-Lipschitz w.r.t. the first argument and bounded by $\frac{c}{c-1}B$.*

*(3) $L_{u_3}(f(\mathbf{x}), \mathbf{y})$ in Eq. (9) is $\sqrt{2}\rho$-Lipschitz w.r.t. the first argument and bounded by $2B$.*

*(4) $L_{u_4}(f(\mathbf{x}), \mathbf{y})$ in Eq. (10) is $\rho\sqrt{c}$-Lipschitz w.r.t. the first argument and bounded by $cB$.*

*Besides, the following holds for the balanced distribution.*

*(1) $L_{pa}(f(\mathbf{x}), \mathbf{y})$ in Eq. (6) is $\frac{\sqrt{2}\rho}{\sqrt{c}}$-Lipschitz w.r.t. the first argument and bounded by $B$.*

*(2) $L_{u_2}(f(\mathbf{x}), \mathbf{y})$ in Eq. (8) is $\frac{4\rho}{c\sqrt{c}}$-Lipschitz w.r.t. the first argument and bounded by $\frac{4}{c}B$.*

*(3) $L_{u_3}(f(\mathbf{x}), \mathbf{y})$ in Eq. (9) is both $\frac{2\rho}{\sqrt{c}}$-Lipschitz w.r.t. the first argument and bounded by $2B$.*

*(4) $L_{u_4}(f(\mathbf{x}), \mathbf{y})$ in Eq. (10) is both $\frac{2\rho}{\sqrt{c}}$-Lipschitz w.r.t. the first argument and bounded by $2B$.*

*Proof.* (a) For the surrogate pairwise loss Eq. (6), $\forall f^1, f^2 \in \mathcal{F}$, the following holds:

$$|L_{pa}(f^1, \mathbf{y}) - L_{pa}(f^2, \mathbf{y})|$$

$$= \left| \frac{1}{|S_{\mathbf{y}}^+||S_{\mathbf{y}}^-|} \sum_{p \in S_{\mathbf{y}}^+} \sum_{q \in S_{\mathbf{y}}^-} \ell(f_p^1 - f_q^1) - \frac{1}{|S_{\mathbf{y}}^+||S_{\mathbf{y}}^-|} \sum_{p \in S_{\mathbf{y}}^+} \sum_{q \in S_{\mathbf{y}}^-} \ell(f_p^2 - f_q^2) \right|$$

$$\leq \frac{1}{|S_{\mathbf{y}}^+||S_{\mathbf{y}}^-|} \sum_{p \in S_{\mathbf{y}}^+} \sum_{q \in S_{\mathbf{y}}^-} |\ell(f_p^1 - f_q^1) - \ell(f_p^2 - f_q^2)| \qquad (|a+b| \leq |a| + |b|)$$

$$\leq \frac{1}{|S_{\mathbf{y}}^+||S_{\mathbf{y}}^-|} \sum_{p \in S_{\mathbf{y}}^+} \sum_{q \in S_{\mathbf{y}}^-} \rho |f_p^1 - f_q^1 - f_p^2 + f_q^2| \qquad (\ell(u) \text{ is } \rho - Lipschitz)$$

$$\leq \rho \left[ \frac{1}{|S_{\mathbf{y}}^+||S_{\mathbf{y}}^-|} \sum_{p \in S_{\mathbf{y}}^+} \sum_{q \in S_{\mathbf{y}}^-} |f_p^1 - f_q^1 - f_p^2 + f_q^2|^2 \right]^{1/2} \qquad (Jense's\ Inequality)$$

$$\leq \rho \left[ \frac{1}{|S_{\mathbf{y}}^+||S_{\mathbf{y}}^-|} \sum_{p \in S_{\mathbf{y}}^+} \sum_{q \in S_{\mathbf{y}}^-} \left\{ |f_p^1 - f_p^2|^2 + |f_q^1 - f_q^2|^2 \right\} \right]^{1/2} \qquad (|a-b|^2 \leq a^2 + b^2)$$

$$= \rho \left[ \frac{1}{|S_{\mathbf{y}}^+||S_{\mathbf{y}}^-|} \left\{ |S_{\mathbf{y}}^-| \sum_{p \in S_{\mathbf{y}}^+} |f_p^1 - f_p^2|^2 + |S_{\mathbf{y}}^+| \sum_{q \in S_{\mathbf{y}}^-} |f_q^1 - f_q^2|^2 \right\} \right]^{1/2}$$

$$\leq \rho \left[ \frac{\max\{|S_{\mathbf{y}}^+|, |S_{\mathbf{y}}^-|\}}{|S_{\mathbf{y}}^+||S_{\mathbf{y}}^-|} \sum_{j=1}^{c} |f_j^1 - f_j^2|^2 \right]^{1/2}$$

$$= \frac{\rho}{\sqrt{\min\{|S_{\mathbf{y}}^+|, |S_{\mathbf{y}}^-|\}}} \|f^1 - f^2\|$$

$$= \heartsuit$$

$$\leq \rho \|f^1 - f^2\| \qquad (1 \leq \min\{|S_{\mathbf{y}}^+|, |S_{\mathbf{y}}^-|\} \leq \frac{c}{2}).$$

As for all distributions, the inequality $1 \leq \min\{|S_{\mathbf{y}}^+|, |S_{\mathbf{y}}^-|\} \leq \frac{c}{2}$ always holds. Next, we consider two extremely cases in the following.

As for extremely imbalanced distributions, we have $\min\{|S_{\mathbf{y}}^+|, |S_{\mathbf{y}}^-|\} = 1$. Then, we have $\heartsuit \leq \rho\|f^1 - f^2\|$.

As for balanced distributions, we have $\min\{|S_{\mathbf{y}}^+|, |S_{\mathbf{y}}^-|\} = \frac{c}{2}$. Then, we have $\heartsuit \leq \frac{\sqrt{2}\rho}{\sqrt{c}}\|f^1 - f^2\|$.

Besides, it is easy to check that $L_{pa}$ is bounded by $B$ for all distributions.

(b) For the surrogate univariate loss $L_{u_2}(f(\mathbf{x}), \mathbf{y}), \forall f^1, f^2 \in \mathcal{F}$, the following holds:

$$
\begin{aligned}
&|L_{u_2}(f^1, \mathbf{y}) - L_{u_2}(f^2, \mathbf{y})| \\
&= \left| \frac{1}{|S_{\mathbf{y}}^+||S_{\mathbf{y}}^-|} \sum_{j=1}^{c} \ell(y_j f_j^1) - \frac{1}{|S_{\mathbf{y}}^+||S_{\mathbf{y}}^-|} \sum_{j=1}^{c} \ell(y_j f_j^2) \right| \\
&\leq \frac{1}{|S_{\mathbf{y}}^+||S_{\mathbf{y}}^-|} \sum_{j=1}^{c} |\ell(y_j f_j^1) - \ell(y_j f_j^2)| && (|a+b| \leq |a| + |b|) \\
&\leq \frac{1}{|S_{\mathbf{y}}^+||S_{\mathbf{y}}^-|} \sum_{j=1}^{c} \rho|y_j f_j^1 - y_j f_j^2| && (\ell(z) \; is \; \rho - Lipschitz) \\
&\leq \frac{\rho c}{|S_{\mathbf{y}}^+||S_{\mathbf{y}}^-|} \left[ \frac{1}{c} \sum_{j=1}^{c} |f_j^1 - f_j^2|^2 \right]^{1/2} && (Jensen's \; Inequality) \\
&= \frac{\rho\sqrt{c}}{|S_{\mathbf{y}}^+||S_{\mathbf{y}}^-|} \|f^1 - f^2\| \\
&= \clubsuit \\
&\leq \frac{\rho\sqrt{c}}{c-1} \|f^1 - f^2\| && (c - 1 \leq |S_{\mathbf{y}}^+||S_{\mathbf{y}}^-| \leq \frac{c^2}{4}).
\end{aligned}
$$

First, as for all distributions, the inequality $c - 1 \leq |S_{\mathbf{y}}^+||S_{\mathbf{y}}^-| \leq \frac{c^2}{4}$ holds. Similarly, we consider two extremely cases in the following.

As for extremely imbalanced distributions, we have $|S_{\mathbf{y}}^+||S_{\mathbf{y}}^-| = c - 1$. Then, we have $\clubsuit \leq \frac{\rho\sqrt{c}}{c-1}\|f^1 - f^2\|$, where is the biggest case for its Lipschitz constant. Besides, it is easy to check that $L_{u_2}$ is bounded by $\frac{c}{c-1}B$.

As for balanced distributions, we have $|S_{\mathbf{y}}^+||S_{\mathbf{y}}^-| = \frac{c^2}{4}$. Then, we have $\clubsuit \leq \frac{4\rho}{c\sqrt{c}}\|f^1 - f^2\|$. Besides, it is easy to check that $L_{u_2}$ is bounded by $\frac{4}{c}B$.

(c) For the surrogate univariate loss $L_{u_3}(f(\mathbf{x}), \mathbf{y}), \forall f^1, f^2 \in \mathcal{F}$, the following holds:

$$
\begin{aligned}
&|L_{u_3}(f^1, \mathbf{y}) - L_{u_3}(f^2, \mathbf{y})| \\
&= \left| \frac{\sum_{p \in S_{\mathbf{y}}^+} [\ell(y_p^1 f_p^1) - \ell(y_p^2 f_p^2)]}{|S_{\mathbf{y}}^+|} + \frac{\sum_{q \in S_{\mathbf{y}}^-} [\ell(y_q^1 f_q^1) - \ell(y_q^2 f_q^2)]}{|S_{\mathbf{y}}^-|} \right| \\
&\leq \frac{\sum_{p \in S_{\mathbf{y}}^+} |\ell(f_p^1) - \ell(f_p^2)|}{|S_{\mathbf{y}}^+|} + \frac{\sum_{q \in S_{\mathbf{y}}^-} |\ell(-f_q^1) - \ell(-f_q^2)|}{|S_{\mathbf{y}}^-|} && (|a+b| \leq |a| + |b|) \\
&\leq \frac{\sum_{p \in S_{\mathbf{y}}^+} \rho|f_p^1 - f_p^2|}{|S_{\mathbf{y}}^+|} + \frac{\sum_{q \in S_{\mathbf{y}}^-} \rho|f_q^1 - f_q^2|}{|S_{\mathbf{y}}^-|} && (\ell(z) \; is \; \rho - Lipschitz) \\
&\leq \rho \left[ \frac{\sum_{p \in S_{\mathbf{y}}^+} |f_p^1 - f_p^2|^2}{|S_{\mathbf{y}}^+|} \right]^{1/2} + \rho \left[ \frac{\sum_{q \in S_{\mathbf{y}}^-} |f_q^1 - f_q^2|^2}{|S_{\mathbf{y}}^-|} \right]^{1/2} && (Jensen's \; Inequality) \\
&\leq \sqrt{2}\rho \left[ \frac{\sum_{p \in S_{\mathbf{y}}^+} |f_p^1 - f_p^2|^2}{|S_{\mathbf{y}}^+|} + \frac{\sum_{q \in S_{\mathbf{y}}^-} |f_q^1 - f_q^2|^2}{|S_{\mathbf{y}}^-|} \right]^{1/2} && (\sqrt{a} + \sqrt{b} \leq \sqrt{2(a+b)})
\end{aligned}
$$

$$\leq \sqrt{2}\rho \left[\frac{\sum_{j=1}^{c} |f_j^1 - f_j^2|^2}{\min\{|S_{\mathbf{y}}^+|, |S_{\mathbf{y}}^-|\}}\right]^{1/2}$$

$$= \bigstar$$

$$\leq \sqrt{2}\rho \|f^1 - f^2\| \qquad\qquad (1 \leq \min\{|S_{\mathbf{y}}^+|, |S_{\mathbf{y}}^-|\} \leq \frac{c}{2}).$$

As for all distributions, $1 \leq \min\{|S_{\mathbf{y}}^+|, |S_{\mathbf{y}}^-|\} \leq \frac{c}{2}$ always holds. Similarly, we consider two extremely cases as follows.

As for extremely imbalanced distributions, we have $\min\{|S_{\mathbf{y}}^+|, |S_{\mathbf{y}}^-|\} = 1$. Then, we have $\bigstar \leq \sqrt{2}\rho\|f^1 - f^2\|$, where is the biggest case for its Lipschitz constant.

As for balanced distributions, we have $\min\{|S_{\mathbf{y}}^+|, |S_{\mathbf{y}}^-|\} = \frac{c}{2}$. Then, we have $\bigstar \leq \frac{2\rho}{\sqrt{c}}\|f^1 - f^2\|$.

Besides, it is easy to check that $L_{u_3}$ is bounded by $2B$ in both cases.

(d) For the surrogate univariate loss $L_{u_4}(f(\mathbf{x}), \mathbf{y}), \forall f^1, f^2 \in \mathcal{F}$, the following holds:

$$|L_{u_4}(f^1, \mathbf{y}) - L_{u_4}(f^2, \mathbf{y})|$$

$$= \left|\frac{1}{\min\{|S_{\mathbf{y}}^+|, |S_{\mathbf{y}}^-|\}}\sum_{j=1}^{c}\ell(y_j f_j^1) - \frac{1}{\min\{|S_{\mathbf{y}}^+|, |S_{\mathbf{y}}^-|\}}\sum_{j=1}^{c}\ell(y_j f_j^2)\right|$$

$$\leq \frac{1}{\min\{|S_{\mathbf{y}}^+|, |S_{\mathbf{y}}^-|\}}\sum_{j=1}^{c}|\ell(y_j f_j^1) - \ell(y_j f_j^2)| \qquad (|a + b| \leq |a| + |b|)$$

$$\leq \frac{1}{\min\{|S_{\mathbf{y}}^+|, |S_{\mathbf{y}}^-|\}}\sum_{j=1}^{c}\rho|y_j f_j^1 - y_j f_j^2| \qquad (\ell(z)\ is\ \rho - Lipschitz)$$

$$\leq \frac{\rho c}{\min\{|S_{\mathbf{y}}^+|, |S_{\mathbf{y}}^-|\}}\left[\frac{1}{c}\sum_{j=1}^{c}|f_j^1 - f_j^2|^2\right]^{1/2} \qquad (Jensen's\ Inequality)$$

$$= \frac{\rho\sqrt{c}}{\min\{|S_{\mathbf{y}}^+|, |S_{\mathbf{y}}^-|\}}\|f^1 - f^2\|$$

$$= \spadesuit$$

$$\leq \rho\sqrt{c}\|f^1 - f^2\| \qquad\qquad (1 \leq \min\{|S_{\mathbf{y}}^+|, |S_{\mathbf{y}}^-|\} \leq \frac{c}{2}).$$

For all distributions, the inequality $1 \leq \min\{|S_{\mathbf{y}}^+|, |S_{\mathbf{y}}^-|\} \leq \frac{c}{2}$ always holds. Similarly, we consider two extremely cases.

As for extremely imbalanced distributions, we have $\min\{|S_{\mathbf{y}}^+|, |S_{\mathbf{y}}^-|\} = 1$. Then, we have $\spadesuit \leq \rho\sqrt{c}\|f^1 - f^2\|$, where is the biggest case for its Lipschitz constant. Besides, it is easy to check that $L_{u_2}$ is bounded by $cB$.

As for balanced distributions, we have $\min\{|S_{\mathbf{y}}^+|, |S_{\mathbf{y}}^-|\} = \frac{c}{2}$. Then, we have $\spadesuit \leq \frac{2\rho}{\sqrt{c}}\|f^1 - f^2\|$. Besides, it is easy to check that $L_{u_2}$ is bounded by $2B$. $\qquad\square$

## B.2 Proof of Lemma 1

**Lemma 1** (The relationship between true and surrogate losses). *For the ranking loss and its surrogate losses, the following inequalities hold:*

$$L_r^{0/1}(f(\mathbf{x}), \mathbf{y}) \leq L_{pa}(f(\mathbf{x}), \mathbf{y}) \leq L_{u_3}(f(\mathbf{x}), \mathbf{y}) \leq L_{u_4}(f(\mathbf{x}), \mathbf{y}) \leq (c - 1)L_{u_2}(f(\mathbf{x}), \mathbf{y}), \quad (4)$$

$$\min\{|S_{\mathbf{y}}^+|, |S_{\mathbf{y}}^-|\}L_{u_2}(f(\mathbf{x}), \mathbf{y}) \leq L_{u_3}(f(\mathbf{x}), \mathbf{y}) \leq \max\{|S_{\mathbf{y}}^+|, |S_{\mathbf{y}}^-|\}L_{u_2}(f(\mathbf{x}), \mathbf{y}). \quad (5)$$

*Besides, note that $L_{u_2}$ cannot strictly upper bound $L_r^{0/1}$ and $L_{pa}$, i.e.,*

$$L_r^{0/1}(f(\mathbf{x}), \mathbf{y}) \not\leq L_{u_2}(f(\mathbf{x}), \mathbf{y}), \ L_{pa}(f(\mathbf{x}), \mathbf{y}) \not\leq L_{u_2}(f(\mathbf{x}), \mathbf{y}).$$

*Proof.* For some widely-used base surrogate loss functions (e.g., the exponential, logistic or hinge loss), it can be easily verified that $\ell(f_p - f_q) \leq \ell(f_p) + \ell(-f_q)$. Thus, the following holds for the first inequality:

$$L_r^{0/1}(f(\mathbf{x}), \mathbf{y}) \leq L_{pa}(f(\mathbf{x}), \mathbf{y})$$

$$= \frac{1}{|S_{\mathbf{y}}^+||S_{\mathbf{y}}^-|} \sum_{(p,q)\in S_{\mathbf{y}}^+ \times S_{\mathbf{y}}^-} \ell(f_p(\mathbf{x}) - f_q(\mathbf{x}))$$

$$\leq \frac{1}{|S_{\mathbf{y}}^+||S_{\mathbf{y}}^-|} \sum_{(p,q)\in S_{\mathbf{y}}^+ \times S_{\mathbf{y}}^-} \left[ \ell(f_p(\mathbf{x})) + \ell(-f_q(\mathbf{x})) \right]$$

$$= \frac{1}{|S_{\mathbf{y}}^+||S_{\mathbf{y}}^-|} \left[ |S_{\mathbf{y}}^-| \sum_{p\in S_{\mathbf{y}}^+} \ell(y_p f_p(\mathbf{x})) + |S_{\mathbf{y}}^+| \sum_{q\in S_{\mathbf{y}}^-} \ell(y_q f_q(\mathbf{x})) \right]$$

$$= L_{u_3}(f(\mathbf{x}), \mathbf{y}).$$

Besides, for the first inequality, the following holds:

$$L_r^{0/1}(f(\mathbf{x}), \mathbf{y}) \leq L_r^{0/1}(sgn \circ f(\mathbf{x}), \mathbf{y})$$

$$= \frac{1}{|S_{\mathbf{y}}^+||S_{\mathbf{y}}^-|} \sum_{p\in S_{\mathbf{y}}^+} \sum_{q\in S_{\mathbf{y}}^-} [\![ sgn(f_p(\mathbf{x})) \leq sgn(f_q(\mathbf{x})) ]\!]$$

$$= \frac{1}{|S_{\mathbf{y}}^+||S_{\mathbf{y}}^-|} \left[ |S_{\mathbf{y}}^-| \sum_{p\in S_{\mathbf{y}}^+} [\![ sgn(f_p(\mathbf{x})) \neq 1 ]\!] + |S_{\mathbf{y}}^+| \sum_{q\in S_{\mathbf{y}}^-} [\![ sgn(f_q(\mathbf{x})) \neq -1 ]\!] - \right.$$

$$\left. \left\{ \sum_{p\in S_{\mathbf{y}}^+} [\![ sgn(f_p(\mathbf{x})) \neq 1 ]\!] \right\} \left\{ \sum_{q\in S_{\mathbf{y}}^-} [\![ sgn(f_q(\mathbf{x})) \neq -1 ]\!] \right\} \right]$$

$$\leq \frac{1}{|S_{\mathbf{y}}^+||S_{\mathbf{y}}^-|} \left[ |S_{\mathbf{y}}^-| \sum_{p\in S_{\mathbf{y}}^+} [\![ sgn(f_p(\mathbf{x})) \neq 1 ]\!] + |S_{\mathbf{y}}^+| \sum_{q\in S_{\mathbf{y}}^-} [\![ sgn(f_q(\mathbf{x})) \neq -1 ]\!] \right]$$

$$= \frac{\sum_{p\in S_{\mathbf{y}}^+} [\![ sgn(f_p(\mathbf{x})) \neq 1 ]\!]}{|S_{\mathbf{y}}^+|} + \frac{\sum_{q\in S_{\mathbf{y}}^-} [\![ sgn(f_q(\mathbf{x})) \neq -1 ]\!]}{|S_{\mathbf{y}}^-|}$$

$$\leq \frac{\sum_{p\in S_{\mathbf{y}}^+} \ell(y_p f_p(\mathbf{x}))}{|S_{\mathbf{y}}^+|} + \frac{\sum_{q\in S_{\mathbf{y}}^-} \ell(y_q f_q(\mathbf{x}))}{|S_{\mathbf{y}}^-|}$$

$$= L_{u_3}(f(\mathbf{x}), \mathbf{y})$$

$$\leq \frac{\max\{|S_{\mathbf{y}}^+|, |S_{\mathbf{y}}^-|\}}{|S_{\mathbf{y}}^+||S_{\mathbf{y}}^-|} \sum_{j=1}^{c} \ell(y_j f_j(\mathbf{x}))$$

$$= \frac{1}{\min\{|S_{\mathbf{y}}^+|, |S_{\mathbf{y}}^-|\}} \sum_{j=1}^{c} \ell(y_j f_j(\mathbf{x}))$$

$$= L_{u_4}(f(\mathbf{x}), \mathbf{y})$$

$$\leq \frac{c-1}{|S_{\mathbf{y}}^+||S_{\mathbf{y}}^-|} \sum_{j=1}^{c} \ell(y_j f_j(\mathbf{x})) \qquad (\frac{c}{2} \leq \max\{|S_{\mathbf{y}}^+|, |S_{\mathbf{y}}^-|\} \leq c-1)$$

$$= (c-1) L_{u_2}(f(\mathbf{x}), \mathbf{y}).$$

Therefore, we can get the first inequality.

For the second inequality, we can get it from the following definitions of $L_{u_2}$ and $L_{u_3}$:

$$L_{u_3} = \frac{\sum_{p\in S_{\mathbf{y}}^+} \ell(y_p f_p(\mathbf{x}))}{|S_{\mathbf{y}}^+|} + \frac{\sum_{q\in S_{\mathbf{y}}^-} \ell(y_q f_q(\mathbf{x}))}{|S_{\mathbf{y}}^-|}$$

$$= \frac{1}{|S_{\mathbf{y}}^+||S_{\mathbf{y}}^-|} \left[ |S_{\mathbf{y}}^-| \sum_{p\in S_{\mathbf{y}}^+} \ell(y_p f_p(\mathbf{x})) + |S_{\mathbf{y}}^+| \sum_{q\in S_{\mathbf{y}}^-} \ell(y_q f_q(\mathbf{x})) \right],$$

$$L_{u_2} = \frac{1}{|S_{\mathbf{y}}^+||S_{\mathbf{y}}^-|} \sum_{j=1}^{c} \ell(y_j f_j(\mathbf{x})).$$

Hence, we can get the inequality:

$$\min\{|S_{\mathbf{y}}^+|, |S_{\mathbf{y}}^-|\} L_{u_2}(f(\mathbf{x}), \mathbf{y}) \leq L_{u_3}(f(\mathbf{x}), \mathbf{y}) \leq \max\{|S_{\mathbf{y}}^+|, |S_{\mathbf{y}}^-|\} L_{u_2}(f(\mathbf{x}), \mathbf{y}).$$

In addition, it is easy to verify that $L_{u_2}$ cannot strictly upper bound $L_r^{0/1}$ and $L_{pa}$, where the proof is omitted here. $\qquad\square$

### B.3 Proofs of Lemma 2 and Theorem 1, 2 and B.2

**Lemma 2** (The relationship between true and surrogate expected risks). *For any $f \in \mathcal{F}$ and any distribution P, the following inequalities hold:*

$$R_{0/1}^{pr}(f) \leq R_{0/1}^r(f) \leq R_{pa}(f) \leq R_{u_3}(f) \leq R_{u_4}(f) \leq (c-1)R_{u_2}(f), \tag{6}$$

$$R_{0/1}^r(f) \not\leq R_{u_2}(f), \quad R_{pa}(f) \not\leq R_{u_2}(f). \tag{7}$$

*Proof.* It is straightforward to apply Lemma 1 to get the results, which is omitted here. $\qquad\square$

Following [4], we also give the base theorem used in the subsequent generalization analysis, as follows.

**Theorem B.1** (The base theorem for generalization analysis [4]). *Assume the loss function $L_\phi : \mathbb{R}^c \times \{-1, +1\}^c \to \mathbb{R}_+$ is $\mu$-Lipschitz continuous w.r.t. the first argument and bounded by $M$. Besides, (1) and (2) in* Assumption 1 *are satisfied. Then, for any $\delta > 0$, with probability at least $1 - \delta$ over the draw of an i.i.d. sample S of size n, the following generalization bound holds for all $f \in \mathcal{F}$:*

$$R_\phi(f) \leq \hat{R}_S(f) + 2\sqrt{2}\mu\sqrt{\frac{c\Lambda^2 r^2}{n}} + 3M\sqrt{\frac{\log \frac{2}{\delta}}{2n}}. \tag{8}$$

#### B.3.1 Proof of Theorem 1

**Theorem 1** (Learning guarantee of $\mathcal{A}^{u_2}$ for extremely imbalanced distribution (worst case)). *Assume the loss $L_\phi = (c-1)L_{u_2}$, where $L_{u_2}$ is defined in Eq. (8). Besides,* Assumption 1 *holds and suppose P is extremely imbalanced. Then, for any $\delta > 0$, with probability at least $1 - \delta$ over S, the following generalization bound holds for all $f \in \mathcal{F}$:*

$$R_{0/1}^r(f) \leq R_{pa}(f) \leq (c-1)R_{u_2}(f) \leq (c-1)\hat{R}_S^{u_2}(f) + 2\sqrt{2}\rho c\sqrt{\frac{\Lambda^2 r^2}{n}} + 3Bc\sqrt{\frac{\log \frac{2}{\delta}}{2n}}. \tag{9}$$

*Proof.* Since $L_\phi = (c-1)L_{u_2}$, we can get its Lipschitz constant (i.e. $\rho\sqrt{c}$) and bounded value (i.e. $cB$) from (2) in Lemma B.1. Then, applying Theorem B.1, we can get that, for any $\delta > 0$, with probability at least $1 - \delta$ over S, the following generalization bound holds for all $f \in \mathcal{F}$:

$$R_\phi(f) = (c-1)R_{u_2}(f) \leq (c-1)\hat{R}_S^{u_2}(f) + 2\sqrt{2}\rho c\sqrt{\frac{\Lambda^2 r^2}{n}} + 3Bc\sqrt{\frac{\log \frac{2}{\delta}}{2n}}. \tag{10}$$

Besides, from Lemma 2, we can get the inequality $R_{0/1}^r(f) \leq R_{pa}(f) \leq (c-1)R_{u_2}(f)$. Thus, we can get this theorem. $\qquad\square$

#### B.3.2 Proof of Theorem 2

**Theorem 2** (Learning guarantee of $\mathcal{A}^{u_3}$ for extremely imbalanced distribution (worst case)). *Assume the loss $L_\phi = L_{u_3}$, where $L_{u_3}$ is defined in Eq. (9). Besides,* Assumption 1 *holds and suppose P is extremely imbalanced. Then, for any $\delta > 0$, with probability at least $1 - \delta$ over S, the following generalization bound holds for all $f \in \mathcal{F}$:*

$$R_{0/1}^r(f) \leq R_{pa}(f) \leq R_{u_3}(f) \leq \hat{R}_S^{u_3}(f) + 4\rho\sqrt{\frac{c\Lambda^2 r^2}{n}} + 6B\sqrt{\frac{\log \frac{2}{\delta}}{2n}}. \tag{11}$$

*Proof.* Since $L_\phi = L_{u_3}$, we can get its Lipschitz constant (i.e. $\sqrt{2}\rho$) and bounded value (i.e. $2B$) from (3) in Lemma B.1. Then, applying Theorem B.1 and the inequality $R_{0/1}^r(f) \leq R_{pa}(f) \leq R_{u_3}(f)$ from Lemma 2, we can get this theorem. $\qquad\square$

### B.3.3 Proof of Theorem B.2

**Theorem B.2** (Learning guarantee of $\mathcal{A}^{u_4}$ for extremely imbalanced distribution (worst case)). *Assume the loss $L_\phi = L_{u_4}$, where $L_{u_4}$ is defined in Eq. (10). Besides,* Assumption 1 *holds and suppose P is extremely imbalanced. Then, for any $\delta > 0$, with probability at least $1 - \delta$ over S, the following generalization bound holds for all $f \in \mathcal{F}$:*

$$R_{0/1}^r(f) \leq R_{pa}(f) \leq R_{u_4}(f) \leq \hat{R}_S^{u_4}(f) + 2\sqrt{2}\rho c \sqrt{\frac{\Lambda^2 r^2}{n}} + 3cB\sqrt{\frac{\log\frac{2}{\delta}}{2n}}. \qquad (12)$$

*Proof.* Since $L_\phi = L_{u_4}$, we can get its Lipschitz constant (i.e. $\rho\sqrt{c}$) and bounded value (i.e. $cB$) from (4) in Lemma B.1. Then, applying Theorem B.1 and the inequality $R_{0/1}^r(f) \leq R_{pa}(f) \leq R_{u_4}(f)$ from Lemma 2, we can get this theorem. $\qquad\square$

## B.4 Proofs of Theorem 3 and B.3

### B.4.1 Proof of Theorem 3

**Theorem 3** (Learning guarantee of $\mathcal{A}^{u_k}, k = 1, 2, 3, 4$ for balanced distribution (best case)). *Assume the loss $L_\phi = 2L_{u_1} = \frac{c}{2}L_{u_2} = L_{u_3} = L_{u_4}$, where they are defined in Section 3.1. Besides,* Assumption 1 *holds and suppose P is balanced. Then, for any $\delta > 0$, with probability at least $1 - \delta$ over S, the following generalization bound holds for all $f \in \mathcal{F}$:*

$$R_{0/1}^r(f) \leq R_{pa}(f) \leq R_{u_3}(f) = \frac{c}{2}R_{u_2}(f) \leq \frac{c}{2}\hat{R}_S^{u_2}(f) + 4\sqrt{2}\rho\sqrt{\frac{\Lambda^2 r^2}{n}} + 6B\sqrt{\frac{\log\frac{2}{\delta}}{2n}}, \qquad (13)$$

*where $2\hat{R}_S^{u_1}(f) = \frac{c}{2}\hat{R}_S^{u_2}(f) = \hat{R}_S^{u_3}(f) = \hat{R}_S^{u_4}(f)$.*

*Proof.* For balanced distributions, it is easy to verify that $2L_{u_1} = \frac{c}{2}L_{u_2} = L_{u_3} = L_{u_4}$ and thus $2R_{u_1}(f) = \frac{c}{2}R_{u_2}(f) = R_{u_3}(f) = R_{u_4}(f)$ and $2\hat{R}_S^{u_1}(f) = \frac{c}{2}\hat{R}_S^{u_2}(f) = \hat{R}_S^{u_3}(f) = \hat{R}_S^{u_4}(f)$. In the following, we take $\mathcal{A}^{u_2}$ for example.

Since $L_\phi = \frac{c}{2}L_{u_2}$, we can get its Lipschitz constant (i.e. $\frac{2\rho}{\sqrt{c}}$) and bounded value (i.e. $2B$) from the balanced case in Lemma B.1. Then, applying Theorem B.1 and the inequality $R_{0/1}^r(f) \leq R_{pa}(f) \leq \frac{c}{2}R_{u_2}(f)$, we can get this theorem.

Similarly, we can get the same learning guarantee for $\mathcal{A}^{u_1}$, $\mathcal{A}^{u_3}$ and $\mathcal{A}^{u_4}$. $\qquad\square$

### B.4.2 Proof of Theorem B.3

**Theorem B.3** (Learning guarantee of $\mathcal{A}^{pa}$ for balanced distribution (best case)). *Assume the loss $L_\phi = L_{pa}$, where $L_{pa}$ is defined in Eq. (6). Besides,* Assumption 1 *holds and suppose P is balanced. Then, for any $\delta > 0$, with probability at least $1 - \delta$ over S, the following generalization bound holds for all $f \in \mathcal{F}$:*

$$R_{0/1}^r(f) \leq R_{pa}(f) \leq \hat{R}_S^{pa}(f) + 4\rho\sqrt{\frac{\Lambda^2 r^2}{n}} + 3B\sqrt{\frac{\log\frac{2}{\delta}}{2n}}. \qquad (14)$$

*Proof.* Since $L_\phi = L_{pa}$, we can get its Lipschitz constant (i.e. $\frac{\sqrt{2}\rho}{\sqrt{c}}$) and bounded value (i.e. $B$) from the balanced case in Lemma B.1. Then, applying Theorem B.1 and the inequality $R_{0/1}^r(f) \leq R_{pa}(f)$ from Lemma 2, we can get this theorem. $\qquad\square$

## B.5 Proofs of Theorem B.5, B.6 and B.7

As for the in-between cases of imbalance, indeed it is highly nontrivial to consider a continuous changing imbalance level of the distribution when instances may have different numbers of positive labels, and we leave it as an important future direction.

Nevertheless, our framework can be applied to the cases where each instance has the same number of positive labels, denoted as $c_p$. $\frac{\min\{c_p, c-c_p\}}{c}$ directly reflects the imbalance level of the distribution. Note that the extremely imbalanced case ($c_p = 1$ or $c_p = c - 1$) and the balanced one ($c_p = c/2$) are included.

First we give the following definition.

**Definition 2** (Instance-wise $c_p$-imbalanced distribution). *For a distribution $P$ for MLC, it is said to be instance-wise $c_p$-imbalanced if for any $(\mathbf{x}, \mathbf{y})$ sampled from $P$, $|S_{\mathbf{y}}^+| = c_p$ always holds.*[2]

For the clarity of following discussions, here we denote $c_{min} = \min\{c_p, c - c_p\}$.

### B.5.1 Proof of Theorem B.5

**Theorem B.5** (Learning guarantee of $\mathcal{A}^{pa}$ for $c_p$-imbalanced distribution). *Assume the loss $L_\phi = L_{pa}$, where $L_{pa}$ is defined in Eq. (6). Besides, Assumption 1 holds and suppose $P$ is $c_p$-imbalanced. Then, for any $\delta > 0$, with probability at least $1 - \delta$ over S, the following generalization bound holds for all $f \in \mathcal{F}$:*

$$R_{0/1}^r(f) \leq R_{pa}(f) \leq \hat{R}_S^{pa}(f) + \frac{2\sqrt{2}\rho}{\sqrt{c_{min}}}\sqrt{\frac{c\Lambda^2 r^2}{n}} + 3B\sqrt{\frac{\log\frac{2}{\delta}}{2n}}. \tag{15}$$

*Proof.* Since $L_\phi = L_{pa}$, we can get its Lipschitz constant (i.e. $\frac{\rho}{\sqrt{c_{min}}}$) and bounded value (i.e. $B$) following the same analysis technique in Lemma B.1. Then, applying Theorem B.1 and the inequality $R_{0/1}^r(f) \leq R_{pa}(f)$ from Lemma 2, we can get this theorem. □

### B.5.2 Proof of Theorem B.6

**Theorem B.6** (Learning guarantee of $\mathcal{A}^{u_2}$ for $c_p$-imbalanced distribution). *Assume the loss $L_\phi = (c - c_{min})L_{u_2}$, where $L_{u_2}$ is defined in Eq. (8). Besides, Assumption 1 holds and suppose $P$ is $c_p$-imbalanced. Then, for any $\delta > 0$, with probability at least $1 - \delta$ over S, the following generalization bound holds for all $f \in \mathcal{F}$:*

$$R_{0/1}^r(f) \leq R_{pa}(f) \leq (c - c_{min})R_{u_2}(f) \leq (c - c_{min})\hat{R}_S^{u_2}(f) + \frac{2\sqrt{2}\rho c}{c_{min}}\sqrt{\frac{\Lambda^2 r^2}{n}} + \frac{3Bc}{c_{min}}\sqrt{\frac{\log\frac{2}{\delta}}{2n}}. \tag{16}$$

*Proof.* Since $L_\phi = (c - c_{min})L_{u_2}$, we can get its Lipschitz constant (i.e. $\frac{\rho\sqrt{c}}{c_{min}(c-c_{min})}$) and bounded value (i.e. $\frac{cB}{c_{min}(c-c_{min})}$) following the same analysis technique in Lemma B.1. Then, applying Theorem B.1, we can get that, for any $\delta > 0$, with probability at least $1 - \delta$ over $S$, the following generalization bound holds for all $f \in \mathcal{F}$:

$$R_\phi(f) = (c - c_{min})R_{u_2}(f) \leq (c - c_{min})\hat{R}_S^{u_2}(f) + \frac{2\sqrt{2}\rho c}{c_{min}}\sqrt{\frac{\Lambda^2 r^2}{n}} + \frac{3Bc}{c_{min}}\sqrt{\frac{\log\frac{2}{\delta}}{2n}}. \tag{17}$$

Besides, based on Lemma 1 and the expected risk definition, we can get the inequality $R_{0/1}^r(f) \leq R_{pa}(f) \leq (c - c_{min})R_{u_2}(f)$. Thus, we can get this theorem. □

---

[2]In this paper we call $c_p$-imbalanced distribution (or dataset) for short.

### B.5.3 Proof of Theorem B.7

**Theorem B.7** (Learning guarantee of $\mathcal{A}^{u_3}$ for $c_p$-imbalanced distribution). *Assume the loss $L_\phi = L_{u_3}$, where $L_{u_3}$ is defined in Eq. (9). Besides,* Assumption 1 *holds and suppose $P$ is $c_p$-imbalanced. Then, for any $\delta > 0$, with probability at least $1 - \delta$ over $S$, the following generalization bound holds for all $f \in \mathcal{F}$:*

$$R_{0/1}^r(f) \leq R_{pa}(f) \leq R_{u_3}(f) \leq \hat{R}_S^{u_3}(f) + \frac{4\rho}{\sqrt{c_{min}}}\sqrt{\frac{c\Lambda^2 r^2}{n}} + 6B\sqrt{\frac{\log \frac{2}{\delta}}{2n}}. \tag{18}$$

*Proof.* Since $L_\phi = L_{u_3}$, we can get its Lipschitz constant (i.e. $\frac{\sqrt{2}\rho}{\sqrt{c_{min}}}$) and bounded value (i.e. $2B$) following the same analysis technique in Lemma B.1. Then, applying Theorem B.1 and the inequality $R_{0/1}^r(f) \leq R_{pa}(f) \leq R_{u_3}(f)$ from Lemma 2, we can get this theorem. $\square$

## C Consistency Analyses

Recall that the ranking loss and the partial ranking loss are defined as

$$L_r^{0/1}(f(\mathbf{x}), \mathbf{y}) = \frac{\sum_{(p,q)\in S_{\mathbf{y}}^+ \times S_{\mathbf{y}}^-} [\![f_p(\mathbf{x}) \leq f_q(\mathbf{x})]\!]}{|S_{\mathbf{y}}^+||S_{\mathbf{y}}^-|}, \tag{19}$$

and

$$L_{pr}^{0/1}(f(\mathbf{x}), \mathbf{y}) = \frac{1}{|S_{\mathbf{y}}^+||S_{\mathbf{y}}^-|} \sum_{(p,q)\in S_{\mathbf{y}}^+ \times S_{\mathbf{y}}^-} \left[ [\![f_p(\mathbf{x}) < f_q(\mathbf{x})]\!] + \frac{1}{2}[\![f_p(\mathbf{x}) = f_q(\mathbf{x})]\!] \right], \tag{20}$$

respectively. For generality, following [3, 2], we do not specify the penalties in the losses at beginning. Recall that the *general ranking loss* is defined as

$$L_{gr}^{0/1}(f(\mathbf{x}), \mathbf{y}) = \alpha_{\mathbf{y}} \sum_{(p,q)\in S_{\mathbf{y}}^+ \times S_{\mathbf{y}}^-} \left[ [\![f_p(\mathbf{x}) \leq f_q(\mathbf{x})]\!] \right], \tag{21}$$

where $\alpha_{\mathbf{y}}$ is a positive penalty, and the *general partial ranking loss* is in a similar form of

$$L_{gpr}^{0/1}(f(\mathbf{x}), \mathbf{y}) = \alpha_{\mathbf{y}} \sum_{(p,q)\in S_{\mathbf{y}}^+ \times S_{\mathbf{y}}^-} \left[ [\![f_p(\mathbf{x}) < f_q(\mathbf{x})]\!] + \frac{1}{2}[\![f_p(\mathbf{x}) = f_q(\mathbf{x})]\!] \right]. \tag{22}$$

The commonly used ranking loss and partial ranking loss are the spacial cases of Eq. (21) and Eq. (22) with $\alpha_{\mathbf{y}} = \frac{1}{|S_{\mathbf{y}}^+||S_{\mathbf{y}}^-|}$ respectively. Also, recall that the *general reweighted univariate surrogate loss* is defined as follows:

$$L_u(f(\mathbf{x}), \mathbf{y}) = \sum_{j=1}^c ([\![y_j = +1]\!]\beta_{\mathbf{y}}^+ + [\![y_j = -1]\!]\beta_{\mathbf{y}}^-)\ell(y_j f_j(\mathbf{x})), \tag{23}$$

where $\beta_{\mathbf{y}}^+$ and $\beta_{\mathbf{y}}^-$ are positive penalties. All univariate surrogate losses mentioned in the main text are spacial cases of Eq. (23), respectively.

Let $\mathcal{B}_L(\mathbf{x}, P(\mathbf{y}|\mathbf{x}))$ denote the set of the Bayes predictors of a loss $L$ given a data point $\mathbf{x}$ and a conditional distribution $P(\mathbf{y}|\mathbf{x})$. Remarkably, a sufficient and necessary condition (called *multi-label consistency* [2]) for a surrogate loss to be (Fisher) consistent w.r.t. the (partial) ranking loss is presented in the following Lemma C.1.

**Lemma C.1** (Multi-label consistency [2]). *A surrogate loss $L$ is consistent w.r.t. a $0/1$ loss $L^{0/1}$, including the general ranking loss in Eq. (21) and the general partial ranking loss in Eq. (22), if and only if $\forall \mathbf{x}$ and $P(\mathbf{y}|\mathbf{x})$, $\mathcal{B}_L(\mathbf{x}, P(\mathbf{y}|\mathbf{x})) \subset \mathcal{B}_{L^{0/1}}(\mathbf{x}, P(\mathbf{y}|\mathbf{x}))$.*

For convenience, we define

$$\Delta_{pq}^{rk} = \sum_{\mathbf{y}:y_p=s_r, y_q=s_k} \alpha_{\mathbf{y}} P(\mathbf{y}|\mathbf{x}), \text{ and } \Delta_p^r = \sum_{\mathbf{y}:y_p=s_r} \alpha_{\mathbf{y}} P(\mathbf{y}|\mathbf{x}) = \Delta_{pq}^{r+} + \Delta_{pq}^{r-}, \forall p \neq q, \tag{24}$$

where $r, k \in \{+, -\}$ and $s_+ = +1$ and $s_- = -1$. The following Lemma C.2 characterizes the set of the Bayes predictors w.r.t. the general ranking loss in Eq. (21) and the general partial ranking loss in Eq. (22).

**Lemma C.2** (Bayes predictor of (partial) ranking loss [2])**.** *For all* $\mathbf{x}$ *and* $P(\mathbf{y}|\mathbf{x})$*, the set of Bayes predictors w.r.t. the general ranking loss in Eq.* (21) *is given by*

$$\mathcal{B}_{L_{gr}^{0/1}}(\mathbf{x}, P(\mathbf{y}|\mathbf{x})) = \{f : \forall 1 \le p < q \le c, f_p > f_q \text{ if } \Delta_{pq}^{+-} > \Delta_{pq}^{-+}; , f_p \ne f_q \text{ if } \Delta_{pq}^{+-} = \Delta_{pq}^{-+}; f_p < f_q \text{ otherwise}\},$$

(25)

*and the set of Bayes predictors w.r.t. the general partial ranking loss in Eq.* (22) *is given by*

$$\mathcal{B}_{L_{gpr}^{0/1}}(\mathbf{x}, P(\mathbf{y}|\mathbf{x})) = \{f : \forall 1 \le p < q \le c, f_p > f_q \text{ if } \Delta_{pq}^{+-} > \Delta_{pq}^{-+}; f_p < f_q \text{ if } \Delta_{pq}^{+-} < \Delta_{pq}^{-+}\}.$$

(26)

Similarly to Eq. (24), we define

$$\phi_p^+ = \sum_{\mathbf{y}:y_p=+1} \beta_{\mathbf{y}}^+ P(\mathbf{y}|\mathbf{x}) \text{ and } \phi_p^- = \sum_{\mathbf{y}:y_p=-1} \beta_{\mathbf{y}}^- P(\mathbf{y}|\mathbf{x}).$$

(27)

The following Lemma C.3 characterizes the set of the Bayes predictors w.r.t. the general reweighted univariate surrogate loss in Eq. (23) with $\ell(z) = e^{-z}$ or $\ell(z) = \ln(1 + e^{-z})$.

**Lemma C.3** (Bayes predictor of Eq. (23) with exponential or logistic loss)**.** *For all* $\mathbf{x}$ *and* $P(\mathbf{y}|\mathbf{x})$*, the set of Bayes predictors w.r.t. the general reweighted univariate surrogate loss in Eq.* (23) *with* $\ell(z) = e^{-z}$ *or* $\ell(z) = \ln(1 + e^{-z})$ *is given by*[3]

$$\mathcal{B}_{L_u}^{\ell}(\mathbf{x}, P(\mathbf{y}|\mathbf{x})) = \{f : \forall 1 \le j \le c, f_j = C \ln \frac{\phi_j^+}{\phi_j^-} \text{ if } \phi_j^+ \phi_j^- > 0; f_j = +\infty \text{ if } \phi_j^- = 0; f_j = -\infty \text{ if } \phi_j^+ = 0\},$$

(28)

*where* $C = \frac{1}{2}$ *if* $\ell(z) = e^{-z}$ *and* $C = 1$ *if* $\ell(z) = \ln(1 + e^{-z})$*.*

The following Lemma C.4 and Lemma C.4 characterize the set of the Bayes predictors w.r.t. the general reweighted univariate surrogate loss in Eq. (23) with $\ell(z) = (\max\{0, 1 - z\})^2$ and $\ell(z) = \max\{0, 1 - z\}$, respectively.

**Lemma C.4** (Bayes predictor of Eq. (23) with squared hinge loss)**.** *For all* $\mathbf{x}$ *and* $P(\mathbf{y}|\mathbf{x})$*, the set of Bayes predictors w.r.t. the general reweighted univariate surrogate loss in Eq.* (23) *with* $\ell(z) = (\max(0, 1 - z))^2$ *is given by*

$$\mathcal{B}_{L_u}^{\ell}(\mathbf{x}, P(\mathbf{y}|\mathbf{x})) = \{f : \forall 1 \le j \le c, f_j = \frac{\phi_j^+ - \phi_j^-}{\phi_j^+ + \phi_j^-}\}.$$

(29)

**Lemma C.4** (Bayes predictor of Eq. (23) with hinge loss)**.** *For all* $\mathbf{x}$ *and* $P(\mathbf{y}|\mathbf{x})$*, the set of Bayes predictors w.r.t. the general reweighted univariate surrogate loss in Eq.* (23) *with* $\ell(z) = \max(0, 1 - z)$ *is given by*

$$\mathcal{B}_{L_u}^{\ell}(\mathbf{x}, P(\mathbf{y}|\mathbf{x})) = \{f : \forall 1 \le j \le c, f_j = 1 \text{ if } \phi_j^+ > \phi_j^-; f_j = -1 \text{ if } \phi_j^+ < \phi_j^-\}.$$

(30)

The proof of Lemma C.3, Lemma C.4 and Lemma C.4 are presented in Appendix C.1. Combining the Lemma C.1, Lemma C.2, Lemma C.3 and Lemma C.4, we have the following sufficient and necessary condition for the general reweighted univariate surrogate loss in Eq. (23) with $\ell(z) = e^{-z}$ or $\ell(z) = \ln(1 + e^{-z})$ or $\ell(z) = (\max\{0, 1 - z\})^2$ to be consistent, as summarized in Proposition C.1.

**Proposition C.1** (Sufficient and necessary condition for the consistency of Eq. (23) w.r.t. Eq. (22) with exponential, logistic or squared hinge loss; proof in Appendix C.2)**.** *The general reweighted univariate surrogate loss in Eq.* (23) *with* $\ell(z) = e^{-z}$ *or* $\ell(z) = \ln(1 + e^{-z})$ *or* $\ell(z) = (\max\{0, 1 - z\})^2$ *is consistent w.r.t. the general partial ranking loss in Eq.* (22) *if and only if for all* $\mathbf{x}$ *and* $P(\mathbf{y}|\mathbf{x})$*, we have*

$$\forall 1 \le p < q \le c, \phi_p^+ \phi_q^- - \phi_p^- \phi_q^+ > 0 \text{ if } \Delta_p^+ \Delta_q^- - \Delta_p^- \Delta_q^+ > 0; \phi_p^+ \phi_q^- - \phi_p^- \phi_q^+ < 0 \text{ if } \Delta_p^+ \Delta_q^- - \Delta_p^- \Delta_q^+ < 0.$$

(31)

---

[3]Because $\sum_{\mathbf{y}} P(\mathbf{y}|\mathbf{x}) = 1$ for any $\mathbf{x}$ and we assume that the penalties are positive, then $\forall 1 \le j \le c$, $\phi_j^+ + \phi_j^- > 0$.

Note that it takes additional efforts to check the consistency of a new surrogate loss according to Lemma C.1 or Proposition C.1, because one has to enumerate all possible conditional distributions. For the general loss in Eq. (23), we present more intuitive characterization that only involves the penalties in Theorem 4 and Proposition 1, considering different base losses.

**Theorem 4** (Necessary condition for the consistency of Eq. (23) w.r.t. Eq. (22) with exponential, logistic or squared hinge loss; proof in Appendix C.3)**.** *A general reweighted univariate surrogate loss in Eq. (23) with $\ell(z) = e^{-z}$, $\ell(z) = \ln(1 + e^{-z})$ or $\ell(z) = (\max\{0, 1 - z\})^2$ is consistent w.r.t. the general partial ranking loss in Eq. (22) only if $\exists \tau > 0$, $\beta_{\mathbf{y}}^+ \beta_{\mathbf{y}}^- = \tau \alpha_{\mathbf{y}}^2$ for all $\mathbf{y}$ such that $1 - c \leq \sum_{1 \leq j \leq c} y_j \leq c - 1$.*

Note that, when $c \leq 3$, the penalties of $L_{u_1}$, $L_{u_3}$ and $L_{u_4}$ may coincide with that of $L_{u_2}$ up to a multiplicative constant. When $c \geq 4$, it is straightforward to construct counter examples that violate the necessary condition in Theorem 4 and obtain the following Corollary 1.

**Corollary 1** (Inconsistency of $L_{u_1}$, $L_{u_3}$ and $L_{u_4}$ w.r.t. Eq. (20) with exponential, logistic or squared hinge loss; proof in Appendix C.4)**.** *If $c \geq 4$, $L_{u_1}$, $L_{u_3}$ and $L_{u_4}$ with $\ell(z) = e^{-z}$ or $\ell(z) = \ln(1 + e^{-z})$ or $\ell(z) = (\max\{0, 1 - z\})^2$ are inconsistent w.r.t. the partial ranking loss in Eq. (20).*

Based on Lemma C.1 and Lemma C.4, we further show the inconsistency of the general reweighted univariate loss in Eq. (23) w.r.t. the general partial ranking loss in Eq. (22) with hinge loss. Note that this includes the inconsistency of $L_{u_1}$, $L_{u_3}$ and $L_{u_4}$ w.r.t. Eq. (20).

**Proposition 1** (Inconsistency of Eq. (23) w.r.t. Eq. (22) with hinge loss; proof in Appendix C.5)**.** *The general reweighted univariate surrogate loss in Eq. (23) with $\ell(z) = \max\{0, 1 - z\}$ are inconsistent w.r.t. the general partial ranking loss in Eq. (22), for all positive penalties $\alpha_{\mathbf{y}}$, $\beta_{\mathbf{y}}^+$ and $\beta_{\mathbf{y}}^-$.*

An immediate conclusion from Corollary 1 and Proposition 1 is that $L_{u_1}$, $L_{u_3}$ and $L_{u_4}$ are inconsistent w.r.t. the ranking loss in Eq. (19) because $\mathcal{B}_{L_r^{0/1}}(\mathbf{x}, P(\mathbf{y}|\mathbf{x})) \subset \mathcal{B}_{L_{pr}^{0/1}}(\mathbf{x}, P(\mathbf{y}|\mathbf{x}))$ [2]. Compared to existing work [3, 2], although Theorem 4 and Proposition 1 are negative, this paper considers surrogate losses in a more general reweighted form, i.e. Eq. (23), which may be of independent interest.

## C.1 Proofs of Lemma C.3, Lemma C.4 and Lemma C.4

According to Eq. (23), the conditional risk for the general reweighted univariate surrogate loss in Eq. (23) is:

$$
\begin{aligned}
R(f|\mathbf{x}) &= \sum_{\mathbf{y}} P(\mathbf{y}|\mathbf{x}) L_u(f(\mathbf{x}), \mathbf{y}) \\
&= \sum_{\mathbf{y}} P(\mathbf{y}|\mathbf{x}) \sum_{j=1}^{c} (\llbracket y_j = +1 \rrbracket \beta_{\mathbf{y}}^+ + \llbracket y_j = -1 \rrbracket \beta_{\mathbf{y}}^-) \ell(y_j f_j), \\
&= \sum_{\mathbf{y}} \sum_{j=1}^{c} (\llbracket y_j = +1 \rrbracket \beta_{\mathbf{y}}^+ + \llbracket y_j = -1 \rrbracket \beta_{\mathbf{y}}^-) P(\mathbf{y}|\mathbf{x}) \ell(y_j f_j), \\
&= \sum_{j=1}^{c} \left[ \sum_{\mathbf{y}: y_j = +1} \beta_{\mathbf{y}}^+ P(\mathbf{y}|\mathbf{x}) \ell(f_j) + \sum_{\mathbf{y}: y_j = -1} \beta_{\mathbf{y}}^- P(\mathbf{y}|\mathbf{x}) \ell(-f_j) \right] \\
&= \sum_{j=1}^{c} \left[ \phi_j^+ \ell(f_j) + \phi_j^- \ell(-f_j) \right].
\end{aligned}
\tag{32}
$$

### C.1.1 Proof of Lemma C.3.

*Proof.* Because $\sum_{\mathbf{y}} P(\mathbf{y}|\mathbf{x}) = 1$ for any $\mathbf{x}$ and we assume that the penalties are positive, then $\forall 1 \leq j \leq c$, $\phi_j^+ + \phi_j^- > 0$. Note that both the exponential loss and logistic loss are strictly monotonically decreasing functions.

According to Eq. (32), if $\phi_j^+ = 0$, then $\phi_j^- \neq 0$ and $f_j^*(\mathbf{x}) = +\infty$. If $\phi_j^- = 0$, then $\phi_j^+ \neq 0$ and $f_j^*(\mathbf{x}) = -\infty$. We now discuss the case where $\phi_j^+ \phi_j^- > 0$.

For the exponential loss $\ell(z) = e^{-z}$, we consider $g(z) = ae^{-z} + be^z$ for $a > 0$ and $b > 0$. It achieves its minima at $z^* = \frac{1}{2} \ln \frac{a}{b}$. To see this, just take the gradient up to the second order and get

$$g'(z) = -ae^{-z} + be^{-z}, g''(z) = ae^{-z} + be^{-z}. \tag{33}$$

Since $\forall z, g''(z) > 0$. Therefore $g(z)$ is convex. Let $g'(z^*) = 0 \Rightarrow z^* = \frac{1}{2} \ln \frac{a}{b}$.

For the logistic loss $\ell(z) = \ln(1 + e^{-z})$, we consider $g(z) = a \ln(1 + e^{-z}) + b \ln(1 + e^z)$ for $a > 0$ and $b > 0$. It achieves its minima at $z^* = \ln \frac{a}{b}$. To see this, just take the gradient up to the second order and get

$$g'(z) = \frac{-ae^{-z}}{1 + e^{-z}} + \frac{be^z}{1 + e^z}, g''(z) = \frac{(a+b)e^z}{(1 + e^z)^2} > 0. \tag{34}$$

Since $\forall z, g''(z) > 0$. Therefore $g(z)$ is convex. Let $g'(z^*) = 0 \Rightarrow z^* = \ln \frac{a}{b}$. Combining all cases together completes the proof. $\qquad \square$

### C.1.2   Proof of Lemma C.4

*Proof.* According to Eq. (32), the conditional risk of the squared hinge loss $\ell(z) = (\max\{0, 1 - z\})^2$ is

$$R(f|\mathbf{x}) = \sum_{j=1}^{c} \left[ \phi_j^+ (\max\{0, 1 - f_j\})^2 + \phi_j^- (\max\{0, 1 + f_j\})^2 \right]. \tag{35}$$

Consider $g(z) = a(\max\{0, 1 - z\})^2 + b(\max\{0, 1 + z\})^2$ for $a \geq 0$, $b \geq 0$ and $a + b > 0$. If $z < -1$, then $g(z) = a(1 - z)^2 > 4a$. If $z > 1$, then $g(z) = b(1 + z)^2 > 4b$. If $-1 \leq z \leq 1$, then $g(z) = (a + b)z^2 + 2(b - a)z + (a + b)$, which is convex. The minima is achieved at $z^* = \frac{a-b}{b+a}$, which satisfies $-1 \leq z^* \leq 1$. The value of $g(z^*)$ is $\frac{4ab}{a+b} \leq \min\{4a, 4b\}$, which means that it is the global minima. Applying this to all $1 \leq j \leq c$ completes the proof. $\qquad \square$

### C.1.3   Proof of Lemma C.4

*Proof.* According to Eq. (32), the conditional risk of the hinge loss $\ell(z) = \max\{0, 1 - z\}$ is

$$R(f|\mathbf{x}) = \sum_{j=1}^{c} \left[ \phi_j^+ \max\{0, 1 - f_j\} + \phi_j^- \max\{0, 1 + f_j\} \right]. \tag{36}$$

Consider $g(z) = a \max\{0, 1 - z\} + b \max\{0, 1 + z\}$ for $a \geq 0$, $b \geq 0$ and $a + b > 0$. If $z < -1$, then $g(z) = a(1 - z) > 2a$. If $z > 1$, then $g(z) = b(1 + z) > 2b$. If $-1 \leq z \leq 1$, then $g(z) = a + b + (b - a)z$. If $b > a$, then $z^* = -1$ and $g(z^*) = 2a < 2b$, which means that it is the global minima. If $b < a$, then $z^* = 1$ and $g(z^*) = 2b < 2a$, which means that it is the global minima. If $b = a$, then whatever $z$ is $g(z) = 2a$. Applying this to all $1 \leq j \leq c$ completes the proof. $\qquad \square$

### C.2   Proof of Proposition C.1

*Proof.* First, note that $\forall p \neq q, \Delta_p^+ + \Delta_p^- = \Delta_q^+ + \Delta_q^- = \sum_{\mathbf{y}} \alpha_{\mathbf{y}} P(\mathbf{y}|\mathbf{x}) > 0$, $\Delta_{pq}^{+-} - \Delta_{pq}^{-+} = \Delta_p^+ - \Delta_q^+$, and

$$\Delta_p^+ \Delta_q^- - \Delta_p^- \Delta_q^+ = \Delta_p^+ \left[ \sum_{\mathbf{y}} \alpha_{\mathbf{y}} P(\mathbf{y}|\mathbf{x}) - \Delta_q^+ \right] - \left[ \sum_{\mathbf{y}} \alpha_{\mathbf{y}} P(\mathbf{y}|\mathbf{x}) - \Delta_p^+ \right] \Delta_q^+ = \left[ \sum_{\mathbf{y}} \alpha_{\mathbf{y}} P(\mathbf{y}|\mathbf{x}) \right] (\Delta_p^+ - \Delta_q^+).$$

Therefore, we have $\forall p < q$,

$$\Delta_{pq}^{+-} > \Delta_{pq}^{-+} \Leftrightarrow \Delta_p^+ > \Delta_q^+ \Leftrightarrow \Delta_p^+ \Delta_q^- - \Delta_p^- \Delta_q^+ > 0,$$

and

$$\Delta_{pq}^{+-} < \Delta_{pq}^{-+} \Leftrightarrow \Delta_p^+ < \Delta_q^+ \Leftrightarrow \Delta_p^+ \Delta_q^- - \Delta_p^- \Delta_q^+ < 0.$$

According to Lemma C.3, when $\ell(z) = e^{-z}$ or $\ell(z) = \ln(1 + e^{-z})$, $\forall f \in \mathcal{B}_{L_u}^{\ell}(\mathbf{x}, P(\mathbf{y}|\mathbf{x}))$, if $\phi_j^+ \phi_j^- > 0$, we have $f_j = C \ln \frac{\phi_j^+}{\phi_j^-}$, where $C$ is a constant. Therefore, $\forall 1 \leq p < q \leq c$, if $\phi_p^+ \phi_p^- > 0$

and $\phi_q^+ \phi_q^- > 0$, then $f_p > f_q \Leftrightarrow \phi_p^+ \phi_q^- - \phi_p^- \phi_q^+ < 0$ and $f_p > f_q \Leftrightarrow \phi_p^+ \phi_q^- - \phi_p^- \phi_q^+ < 0$. It is easy to check this also holds if $\phi_p^+ \phi_q^- = 0$ or $\phi_q^+ \phi_q^- = 0$. Note that we do not need to consider the cases where $\phi_p^+ = \phi_q^+ = 0$ or $\phi_p^- = \phi_q^- = 0$ because they imply $\Delta_p^+ \Delta_q^- - \Delta_p^- \Delta_q^+ = 0$. Combining with Lemma C.1, we complete the proof for the logistic loss and exponential loss.

According to Lemma C.4, when $\ell(z) = (\max\{0, 1-z\})^2$, $\forall f \in \mathcal{B}_{L_u}^\ell (\mathbf{x}, P(\mathbf{y}|\mathbf{x}))$, $1 \leq j \leq c$, $f_j = \frac{\phi_j^+ - \phi_j^-}{\phi_j^+ + \phi_j^-}$. Therefore, $\forall 1 \leq p < q \leq c$, $f_p > f_q \Leftrightarrow \frac{\phi_p^+ - \phi_p^-}{\phi_p^+ + \phi_p^-} > \frac{\phi_q^+ - \phi_q^-}{\phi_q^+ + \phi_q^-} \Leftrightarrow \phi_p^+ \phi_q^- - \phi_p^- \phi_q^+ < 0$ and $f_p > f_q \Leftrightarrow \frac{\phi_p^+ - \phi_p^-}{\phi_p^+ + \phi_p^-} < \frac{\phi_q^+ - \phi_q^-}{\phi_q^+ + \phi_q^-} \Leftrightarrow \phi_p^+ \phi_q^- - \phi_p^- \phi_q^+ < 0$. Combining with Lemma C.1, we complete the proof for the squared hinge loss.

$\square$

### C.3 Proof of Theorem 4

*Proof.* For convenience, for all $p \neq q$, we define

$$\phi_{pq}^{rk} = \sum_{\mathbf{y}: y_p = s_r, y_q = s_k} ([\![y_p = +1]\!]\beta_{\mathbf{y}}^+ + [\![y_p = -1]\!]\beta_{\mathbf{y}}^-) P(\mathbf{y}|\mathbf{x}), \tag{37}$$

where $r, k \in \{+, -\}$ and $s_+ = +1$ and $s_- = -1$. Note that for all $p \neq q$, $\phi_{pq}^{++} = \phi_{qp}^{++}$ and $\phi_{pq}^{--} = \phi_{qp}^{--}$ according to the definition. For all $1 \leq p < q \leq c$, we have

$$\phi_p^+ \phi_q^- - \phi_p^- \phi_q^+ = (\phi_{pq}^{++} + \phi_{pq}^{+-})(\phi_{qp}^{-+} + \phi_{qp}^{--}) - (\phi_{pq}^{-+} + \phi_{pq}^{--})(\phi_{qp}^{++} + \phi_{qp}^{+-})$$
$$= \phi_{pq}^{++} \phi_{qp}^{-+} + \phi_{pq}^{+-} \phi_{qp}^{-+} + \phi_{pq}^{+-} \phi_{qp}^{--} - \phi_{pq}^{-+} \phi_{qp}^{+-} - \phi_{pq}^{-+} \phi_{qp}^{++} - \phi_{pq}^{--} \phi_{qp}^{+-}, \tag{38}$$

and similarly

$$\Delta_p^+ \Delta_q^- - \Delta_p^- \Delta_q^+ = (\Delta_{pq}^{++} + \Delta_{pq}^{+-})(\Delta_{qp}^{-+} + \Delta_{qp}^{--}) - (\Delta_{pq}^{-+} + \Delta_{pq}^{--})(\Delta_{qp}^{++} + \Delta_{qp}^{+-})$$
$$= \Delta_{pq}^{++} \Delta_{qp}^{-+} + \Delta_{pq}^{+-} \Delta_{qp}^{-+} + \Delta_{pq}^{+-} \Delta_{qp}^{--} - \Delta_{pq}^{-+} \Delta_{qp}^{+-} - \Delta_{pq}^{-+} \Delta_{qp}^{++} - \Delta_{pq}^{--} \Delta_{qp}^{+-}. \tag{39}$$

For simplicity, we say a $\mathbf{y}$ is *nontrivial* if it satisfies $1 - c \leq \sum_{1 \leq j \leq c} y_j \leq c - 1$. Assume the consistency holds. We prove that $\exists \tau > 0$, $\beta_{\mathbf{y}}^+ \beta_{\mathbf{y}}^- = \tau \alpha_{\mathbf{y}}^2$ for all nontrivial $\mathbf{y}$. The proof consists of two main steps.

**Step 1:** We first prove that, for all $1 \leq p < q \leq c$, there exists $\tau > 0$, $\beta_{\mathbf{y}}^+ \beta_{\mathbf{y}}^- = \tau \alpha_{\mathbf{y}}^2$ for all $\mathbf{y}$ such that $y_p y_q = -1$. According to Proposition C.1, $\forall \mathbf{x}$ and $P(\mathbf{y}|\mathbf{x})$,

$$\forall p < q, \phi_p^+ \phi_q^- - \phi_p^- \phi_q^+ > 0 \text{ if } \Delta_p^+ \Delta_q^- - \Delta_p^- \Delta_q^+ > 0; \phi_p^+ \phi_q^- - \phi_p^- \phi_q^+ < 0 \text{ if } \Delta_p^+ \Delta_q^- - \Delta_p^- \Delta_q^+ < 0.$$

We simply consider the cases where $P(\mathbf{y}|\mathbf{x}) = 0$ for all $\mathbf{y}$ such that $y_p = y_q$. According to Eq. (38) and Eq. (39), we get

$$\phi_p^+ \phi_q^- - \phi_p^- \phi_q^+ = \phi_{pq}^{+-} \phi_{qp}^{-+} - \phi_{pq}^{-+} \phi_{qp}^{+-}$$
$$= \sum_{\substack{\mathbf{y}: y_p = +1, y_q = -1 \\ \mathbf{y}': y_p' = +1, y_q' = -1}} \beta_{\mathbf{y}}^+ \beta_{\mathbf{y}'}^- P(\mathbf{y}|\mathbf{x}) P(\mathbf{y}'|\mathbf{x}) - \sum_{\substack{\mathbf{y}: y_p = -1, y_q = +1 \\ \mathbf{y}': y_p' = -1, y_q' = +1}} \beta_{\mathbf{y}}^+ \beta_{\mathbf{y}'}^- P(\mathbf{y}|\mathbf{x}) P(\mathbf{y}'|\mathbf{x}), \tag{40}$$

and

$$\Delta_p^+ \Delta_q^- - \Delta_p^- \Delta_q^+ = \Delta_{pq}^{+-} \Delta_{qp}^{-+} - \Delta_{pq}^{-+} \Delta_{qp}^{+-}$$
$$= \sum_{\substack{\mathbf{y}: y_p = +1, y_q = -1 \\ \mathbf{y}': y_p' = +1, y_q' = -1}} \alpha_{\mathbf{y}} \alpha_{\mathbf{y}'} P(\mathbf{y}|\mathbf{x}) P(\mathbf{y}'|\mathbf{x}) - \sum_{\substack{\mathbf{y}: y_p = -1, y_q = +1 \\ \mathbf{y}': y_p' = -1, y_q' = +1}} \alpha_{\mathbf{y}} \alpha_{\mathbf{y}'} P(\mathbf{y}|\mathbf{x}) P(\mathbf{y}'|\mathbf{x}). \tag{41}$$

We proceed by contradiction and consider two cases. Recall that we assume $\alpha_{\mathbf{y}} > 0$ and $\beta_{\mathbf{y}}^+ \beta_{\mathbf{y}}^- > 0$ for all nontrivial $\mathbf{y}$. Suppose that there exists $\tau_3 > 0, \tau_4 > 0, \tau_3 \neq \tau_4$, $\beta_{\mathbf{y}}^+ \beta_{\mathbf{y}}^- = \tau_3 \alpha_{\mathbf{y}}^2 \neq \tau_4 \alpha_{\mathbf{y}}^2$ and $\beta_{\mathbf{y}'}^+ \beta_{\mathbf{y}'}^- = \tau_4 \alpha_{\mathbf{y}'}^2 \neq \tau_3 \alpha_{\mathbf{y}'}^2$ for some $\mathbf{y} \neq \mathbf{y}'$ with $y_p y_q = -1$ and $y_p' y_q' = -1$.

**Case 1.1:** $y_p \neq y'_p$. Without loss of generality, let $y_p = +1$ and $y'_p = -1$. Accordingly, we get $y_q = -1$ and $y'_q = +1$. Let $P(\mathbf{y}|\mathbf{x}) = \frac{\sqrt{\beta^+_{\mathbf{y}'}\beta^-_{\mathbf{y}'}}}{\sqrt{\beta^+_{\mathbf{y}}\beta^-_{\mathbf{y}}}+\sqrt{\beta^+_{\mathbf{y}'}\beta^-_{\mathbf{y}'}}}$ and $P(\mathbf{y}'|\mathbf{x}) = \frac{\sqrt{\beta^+_{\mathbf{y}}\beta^-_{\mathbf{y}}}}{\sqrt{\beta^+_{\mathbf{y}}\beta^-_{\mathbf{y}}}+\sqrt{\beta^+_{\mathbf{y}'}\beta^-_{\mathbf{y}'}}}$. Note that $P(\mathbf{y}|\mathbf{x}) + P(\mathbf{y}'|\mathbf{x}) = 1$. According to Eq. (41), we have

$$\Delta^+_p \Delta^-_q - \Delta^-_p \Delta^+_q = \alpha^2_{\mathbf{y}} P(\mathbf{y}|\mathbf{x})^2 - \alpha^2_{\mathbf{y}'} P(\mathbf{y}'|\mathbf{x})^2 = \frac{\beta^+_{\mathbf{y}}\beta^-_{\mathbf{y}}\beta^+_{\mathbf{y}'}\beta^-_{\mathbf{y}'}}{(\sqrt{\beta^+_{\mathbf{y}}\beta^-_{\mathbf{y}}}+\sqrt{\beta^+_{\mathbf{y}'}\beta^-_{\mathbf{y}'}})^2}(\frac{1}{\tau_4} - \frac{1}{\tau_3}) \neq 0,$$

but according to Eq. (40), we have

$$\phi^+_p \phi^-_q - \phi^-_p \phi^+_q = \beta^+_{\mathbf{y}}\beta^-_{\mathbf{y}} P(\mathbf{y}|\mathbf{x})^2 - \beta^+_{\mathbf{y}'}\beta^-_{\mathbf{y}'} P(\mathbf{y}'|\mathbf{x})^2 = 0,$$

which is a contrary to Proposition C.1.

**Case 1.2:** $y_p = y'_p$. Without loss of generality, let $y_p = y'_p = -1$, then $y_q = y'_q = +1$. Consider $y''_p$ such that $y''_p = +1, y''_q = -1$. Then according to the Case 1.1, there exists $\tau$, such that $\tau = \frac{\beta^+_{\mathbf{y}}\beta^-_{\mathbf{y}}}{\alpha^2_{\mathbf{y}}} = \frac{\beta^+_{\mathbf{y}''}\beta^-_{\mathbf{y}''}}{\alpha^2_{\mathbf{y}''}} = \frac{\beta^+_{\mathbf{y}'}\beta^-_{\mathbf{y}'}}{\alpha^2_{\mathbf{y}'}}$, which is a contrary.

Combining the Case 1.1 and Case 1.2 together, for all $1 \leq p < q \leq c$, there exists $\tau > 0$, $\beta^+_{\mathbf{y}}\beta^-_{\mathbf{y}} = \tau\alpha^2_{\mathbf{y}}$ for all $\mathbf{y}$ such that $y_p y_q = -1$.

**Step 2:** Note that the values of $\tau$ in Step 1 may depend on $p$ and $q$. Now we prove that there exists a universal $\tau$ for all $1 \leq p < q \leq c$. For any nontrivial $\mathbf{y} \neq \mathbf{y}'$, we can find $1 \leq p < q \leq c$ and $1 \leq p' < q' \leq c$ such that $y_p y_q = -1$ and $y'_{p'} y'_{q'} = -1$. We consider four cases.

**Case 2.1:** Two pair of indices match, namely, $p = p'$, $q = q'$. We have proven that $\frac{\beta^-_{\mathbf{y}}\beta^+_{\mathbf{y}}}{\alpha^2_{\mathbf{y}}} = \frac{\beta^-_{\mathbf{y}'}\beta^+_{\mathbf{y}'}}{\alpha^2_{\mathbf{y}'}}$ in Step 1.

**Case 2.2:** No index matches for $c \geq 4$, namely, $p \neq p'$, $q \neq q'$, $p \neq q'$, $p' \neq q$. We can construct $\mathbf{y}''$ such that $y''_p = y_p, y''_q = y_q, y''_{p'} = y'_{p'}, y''_{q'} = y'_{q'}$ and get $\frac{\beta^-_{\mathbf{y}}\beta^+_{\mathbf{y}}}{\alpha^2_{\mathbf{y}}} = \frac{\beta^-_{\mathbf{y}''}\beta^+_{\mathbf{y}''}}{\alpha^2_{\mathbf{y}''}} = \frac{\beta^-_{\mathbf{y}'}\beta^+_{\mathbf{y}'}}{\alpha^2_{\mathbf{y}'}}$ according to Step 1.

**Case 2.3:** Only one pair of indices match and the corresponding labels are the same for $c \geq 3$. Without loss of generality, suppose $p = 1$, $q = p' = 2$, $q' = 3$ and $y_2 = +1, y'_2 = +1$. It implies that $y_1 = -1$ and $y'_3 = -1$. Suppose $y_3 = -1$, then $y_2 y_3 = y'_2 y'_3 = -1$. Suppose $y_3 = +1$, no matter which label $y'_1$ is, either $y_1 y_2 = y'_1 y'_2 = -1$ or $y_1 y_3 = y'_1 y'_3 = -1$. We get $\frac{\beta^-_{\mathbf{y}}\beta^+_{\mathbf{y}}}{\alpha^2_{\mathbf{y}}} = \frac{\beta^-_{\mathbf{y}'}\beta^+_{\mathbf{y}'}}{\alpha^2_{\mathbf{y}'}}$ according to Step 1.

**Case 2.4:** Only one pair of indices match and the corresponding labels are not the same for $c \geq 3$. Without loss of generality, suppose $p = 1$, $q = p' = 2$, and $q' = 3$ and $y_2 = +1, y'_2 = -1$. We have $y_1 = -1, y'_3 = +1$. Similarly to Case 3, no matter which labels $y_3$ and $y'_1$ are, we have either $y_1 y_2 = y'_1 y'_2 = -1$ or $y_1 y_3 = y'_1 y'_3 = -1$ or $y_2 y_3 = y'_2 y'_3 = -1$, and get $\frac{\beta^-_{\mathbf{y}}\beta^+_{\mathbf{y}}}{\alpha^2_{\mathbf{y}}} = \frac{\beta^-_{\mathbf{y}'}\beta^+_{\mathbf{y}'}}{\alpha^2_{\mathbf{y}'}}$ according to Step 1.

Combining Case 2.1, Case 2.2, Case 2.3 and Case 2.4 together, we obtain that for all nontrivial $\mathbf{y} \neq \mathbf{y}'$, $\frac{\beta^-_{\mathbf{y}}\beta^+_{\mathbf{y}}}{\alpha^2_{\mathbf{y}}} = \frac{\beta^-_{\mathbf{y}'}\beta^+_{\mathbf{y}'}}{\alpha^2_{\mathbf{y}'}}$. $\qquad\square$

### C.4 Proof of Corollary 1

*Proof.* We consider a multi-label classification problem with $c \geq 4$ labels. Let $\mathbf{y}$ satisfy $y_1 = +1$, and $y_j = -1$ for all $2 \leq j \leq c$, and $\mathbf{y}'$ satisfy $y'_1 = y'_2 = +1$, and $y'_j = -1$ for all $3 \leq j \leq c$. According to the definition of the partial ranking loss in Eq. (20), we have $\alpha_{\mathbf{y}} = \frac{1}{c-1}$ and $\alpha_{\mathbf{y}'} = \frac{1}{2(c-2)}$.

In $L_{u_1}$, according to the definition, we have $\beta^+_{\mathbf{y}} = \beta^+_{\mathbf{y}'} = \beta^-_{\hat{\mathbf{y}}} = \beta^-_{\mathbf{y}'} = \frac{1}{c}$. It is easy to check that $\frac{\beta^+_{\mathbf{y}}\beta^-_{\mathbf{y}}}{\alpha^2_{\mathbf{y}}} = \frac{(c-1)^2}{c^2} \neq \frac{4(c-2)^2}{c^2} = \frac{\beta^+_{\mathbf{y}'}\beta^-_{\mathbf{y}'}}{\alpha^2_{\mathbf{y}'}}$ for all $c \geq 4$.

In $L_{u_3}$, according to the definition, we have $\beta_{\mathbf{y}}^+ = 1$, $\beta_{\mathbf{y}'}^+ = \frac{1}{2}$, $\beta_{\mathbf{y}}^- = \frac{1}{c-1}$, and $\beta_{\mathbf{y}'}^- = \frac{1}{c-2}$. It is easy to check that $\frac{\beta_{\mathbf{y}}^+ \beta_{\mathbf{y}}^-}{\alpha_{\mathbf{y}}^2} = c - 1 \neq 2(c-2) = \frac{\beta_{\mathbf{y}'}^+ \beta_{\mathbf{y}'}^-}{\alpha_{\mathbf{y}'}^2}$ for all $c \geq 4$.

In $L_{u_4}$, according to the definition, we have $\beta_{\mathbf{y}}^+ = 1$, $\beta_{\mathbf{y}'}^+ = \frac{1}{2}$, $\beta_{\mathbf{y}}^- = 1$, and $\beta_{\mathbf{y}'}^- = \frac{1}{2}$, for all $c \geq 4$. It is easy to check that $\frac{\beta_{\mathbf{y}}^+ \beta_{\mathbf{y}}^-}{\alpha_{\mathbf{y}}^2} = (c-1)^2 \neq (c-2)^2 = \frac{\beta_{\mathbf{y}'}^+ \beta_{\mathbf{y}'}^-}{\alpha_{\mathbf{y}'}^2}$.

According to Theorem 4 and Proposition C.1, the above surrogate losses are not consistent w.r.t. the partial ranking loss in Eq. (20). $\qquad\square$

### C.5 Proof of Proposition 1

*Proof.* We consider a multi-label classification problem with $c = 2$ labels. Let

$$\mathbf{y}_1 = (+1, +1), \mathbf{y}_2 = (+1, -1), \mathbf{y}_3 = (+1, -1), \mathbf{y}_4 = (-1, -1).$$

Given a data point $\mathbf{x}$, let $0 < \epsilon < \frac{\beta_{\mathbf{y}_1}^+}{\beta_{\mathbf{y}_1}^+ + \max\{\beta_{\mathbf{y}_2}^-, \beta_{\mathbf{y}_3}^-\}}$. Consider a conditional distribution such that $P(\mathbf{y}_2|\mathbf{x})P(\mathbf{y}_3|\mathbf{x}) > 0$, $\alpha_{\mathbf{y}_2}P(\mathbf{y}_2|\mathbf{x}) \neq \alpha_{\mathbf{y}_3}P(\mathbf{y}_3|\mathbf{x})$, $P(\mathbf{y}_2|\mathbf{x}) + P(\mathbf{y}_3|\mathbf{x}) = \epsilon$, $P(\mathbf{y}_1|\mathbf{x}) = 1 - \epsilon$ and $P(\mathbf{y}_4|\mathbf{x}) = 0$. On one hand, we get

$$\Delta_1^+ - \Delta_2^+ = \alpha_{\mathbf{y}_2}P(\mathbf{y}_2|\mathbf{x}) - \alpha_{\mathbf{y}_3}P(\mathbf{y}_3|\mathbf{x}) \neq 0, \tag{42}$$

which implies $f_1 \neq f_2$ for any $f \in \mathcal{B}_{L_{gpr}^{0/1}}(\mathbf{x}, P(\mathbf{y}|\mathbf{x}))$ according to Lemma C.2. On the other hand, we get

$$
\begin{aligned}
\phi_1^+ - \phi_1^- &= \beta_{\mathbf{y}_1}^+ P(\mathbf{y}_1|\mathbf{x}) + \beta_{\mathbf{y}_2}^+ P(\mathbf{y}_2|\mathbf{x}) - \beta_{\mathbf{y}_3}^- P(\mathbf{y}_3|\mathbf{x}) - \beta_{\mathbf{y}_4}^- P(\mathbf{y}_4|\mathbf{x}) \\
&> \beta_{\mathbf{y}_1}^+ P(\mathbf{y}_1|\mathbf{x}) - \beta_{\mathbf{y}_3}^- P(\mathbf{y}_3|\mathbf{x}) \\
&> \beta_{\mathbf{y}_1}^+ (1 - \epsilon) - \beta_{\mathbf{y}_3}^- \epsilon \\
&= \beta_{\mathbf{y}_1}^+ \left(1 - \frac{\beta_{\mathbf{y}_1}^+}{\beta_{\mathbf{y}_1}^+ + \max\{\beta_{\mathbf{y}_2}^-, \beta_{\mathbf{y}_3}^-\}}\right) - \beta_{\mathbf{y}_3}^- \frac{\beta_{\mathbf{y}_1}^+}{\beta_{\mathbf{y}_1}^+ + \max\{\beta_{\mathbf{y}_3}^-, \beta_{\mathbf{y}_3}^-\}} \\
&= \frac{\beta_{\mathbf{y}_1}^+ (\max\{\beta_{\mathbf{y}_2}^-, \beta_{\mathbf{y}_3}^-\} - \beta_{\mathbf{y}_3}^-)}{\beta_{\mathbf{y}_1}^+ + \max\{\beta_{\mathbf{y}_2}^-, \beta_{\mathbf{y}_3}^-\}} \\
&\geq 0, \tag{43}
\end{aligned}
$$

which means that $\forall f \in \mathcal{B}_{L_u}^{\ell}(\mathbf{x}, P(\mathbf{y}|\mathbf{x}))$, $f_1 = -1$ according to Lemma C.4. Similarly, $\forall f \in \mathcal{B}_{L_u}(\mathbf{x}, P(\mathbf{y}|\mathbf{x}))$, $f_2 = -1 = f_1$. Therefore, $\mathcal{B}_{L_u}^{\ell}(\mathbf{x}, P(\mathbf{y}|\mathbf{x})) \not\subset \mathcal{B}_{L^{0/1}}(\mathbf{x}, P(\mathbf{y}|\mathbf{x}))$, which completes the proof combining with Lemma C.1. $\qquad\square$

## D  Dataset Details

The detailed statistics of the used dataset is given in Table 1. These datasets can be downloaded from http://mulan.sourceforge.net/datasets-mlc.html and http://palm.seu.edu.cn/zhangml/

## E  Additional Experimental Results

The complete experimental results (with standard deviations) are summarized in Table 2.

Besides, the computational costs of all five algorithms on benchmark datasets are shown in Fig. 1. From Fig. 1, we can observe that $\mathcal{A}^{pa}$ with the pairwise loss is much slower than the other four algorithms with the univariate loss, especially when the label space is large. Note that the CPU time is plotted in the log scale in Figure 1.

We illustrate the instance-wise class imbalances of the benchmark datasets in Fig. 2. From Fig.2, we can observe that the real datasets are highly imbalanced, which is similar to the extremely imbalanced case.

Table 1: Basic statistics of the benchmark datasets.

| Dataset | #Instance | #Feature | #Label | Domain |
|---------|-----------|----------|--------|--------|
| emotions | 593 | 72 | 6 | music |
| image | 2000 | 294 | 5 | images |
| scene | 2407 | 294 | 6 | images |
| yeast | 2417 | 103 | 14 | biology |
| enron | 1702 | 1001 | 53 | text |
| rcv1-subset1 | 6000 | 944 | 101 | text |
| bibtex | 7395 | 1836 | 159 | text |
| corel5k | 5000 | 499 | 374 | images |
| mediamill | 43907 | 120 | 101 | video |
| delicious | 16105 | 500 | 983 | text(web) |

Table 2: Ranking loss (mean $\pm$ std) of all five algorithms on benchmark datasets. On each dataset, the top two algorithms are highlighted in bold and the top one is labeled with $\dagger$.

| Dataset | $\mathcal{A}^{pa}$ | $\mathcal{A}^{u_1}$ | $\mathcal{A}^{u_2}$ | $\mathcal{A}^{u_3}$ | $\mathcal{A}^{u_4}$ |
|---------|--------------------|--------------------|--------------------|--------------------|--------------------|
| emotions | $\mathbf{0.1511 \pm 0.0175}^{\dagger}$ | $0.1538 \pm 0.0219$ | $0.1587 \pm 0.0198$ | $\mathbf{0.1530 \pm 0.0193}$ | $0.1616 \pm 0.0202$ |
| image | $\mathbf{0.1625 \pm 0.0089}^{\dagger}$ | $0.1642 \pm 0.0132$ | $0.1653 \pm 0.0153$ | $\mathbf{0.1645 \pm 0.0159}$ | $0.1678 \pm 0.0056$ |
| scene | $\mathbf{0.0696 \pm 0.0031}^{\dagger}$ | $0.0809 \pm 0.0083$ | $0.0821 \pm 0.0029$ | $\mathbf{0.0768 \pm 0.0082}$ | $0.0806 \pm 0.0025$ |
| yeast | $\mathbf{0.1766 \pm 0.0078}^{\dagger}$ | $0.1768 \pm 0.0093$ | $0.1785 \pm 0.0090$ | $\mathbf{0.1767 \pm 0.0086}$ | $0.1816 \pm 0.0084$ |
| enron | $\mathbf{0.0682 \pm 0.0030}^{\dagger}$ | $0.0724 \pm 0.0022$ | $\mathbf{0.0696 \pm 0.0011}$ | $0.0698 \pm 0.0027$ | $0.0715 \pm 0.0038$ |
| rcv1-subset1 | $\mathbf{0.0361 \pm 0.0015}^{\dagger}$ | $0.0418 \pm 0.0005$ | $0.0392 \pm 0.0003$ | $\mathbf{0.0368 \pm 0.0003}$ | $0.0391 \pm 0.0005$ |
| bibtex | $\mathbf{0.0516 \pm 0.0014}$ | $0.0545 \pm 0.0018$ | $0.0551 \pm 0.0024$ | $\mathbf{0.0401 \pm 0.0694}^{\dagger}$ | $0.0538 \pm 0.0020$ |
| corel5k | $\mathbf{0.1081 \pm 0.0021}$ | $0.1091 \pm 0.0004$ | $0.1099 \pm 0.0016$ | $\mathbf{0.1063 \pm 0.0019}^{\dagger}$ | $0.1096 \pm 0.0010$ |
| mediamill | $\mathbf{0.0395 \pm 0.0011}$ | $0.0402 \pm 0.0005$ | $0.0412 \pm 0.0001$ | $\mathbf{0.0389 \pm 0.0006}^{\dagger}$ | $0.0405 \pm 0.0010$ |
| delicious | - | $\mathbf{0.0960 \pm 0.0010}$ | $0.0974 \pm 0.0007$ | $\mathbf{0.0946 \pm 0.0002}^{\dagger}$ | $0.0978 \pm 0.0008$ |

To further study the effect of the label size (i.e. $c$), we conduct experiments with $\mathcal{A}^{u_2}$ and $\mathcal{A}^{u_3}$ on the semi-synthetic datasets with randomly selected $c$ on the delicious datasets. The imbalances of the semi-synthetic datasets are shown in Fig. 4 and the experimental results are illustrated in Fig. 3. Then, we can observe that $\mathcal{A}^{u_3}$ would probably perform better than $\mathcal{A}^{u_2}$ when the label size $c$ is larger, which confirms our theoretical findings.

Furthermore, to study whether the upper bound for the generalization error can reflect on the true generalization error reasonably well, we conduct experiments on the semi-synthetic delicious datasets, where the results are summarized in Table 3. From Table 3, we can have the following observations.

- On one hand, the surrogate expected pairwise risk $R_{pa}$ can usually reflect the true (or $0/1$) expected risk $R_{0/1}^r$ reasonably well. Further, the tighter $R_{pa}$ and PUB (i.e., the probabilistic upper bound) are usually associated with better $R_{0/1}^r$.[4] This is because that the tighter PUB, which is allowed to be bigger than 1, probably suggests tighter $R_{pa}$, which usually indicates better $R_{0/1}^r$.

- On the other hand, we can observe that the PUB values are somehow large, which are bigger than 1 and might not reflect the expected risk $R_{0/1}^r$ reasonably well. Despite this limitation, it can still offer insights into these learning algorithms — when comparing algorithms, tighter PUB probably suggests better performance and it might be more reasonable to compare the order of dependent variables rather than the absolute values. Besides, advanced techniques (e.g., local Rademacher complexity [5]) might provide more reasonable PUB.

---

[4]Note that we use the error bounds for the worst case which holds on all distribution. Besides, it might be better to employ the distribution dependent one.

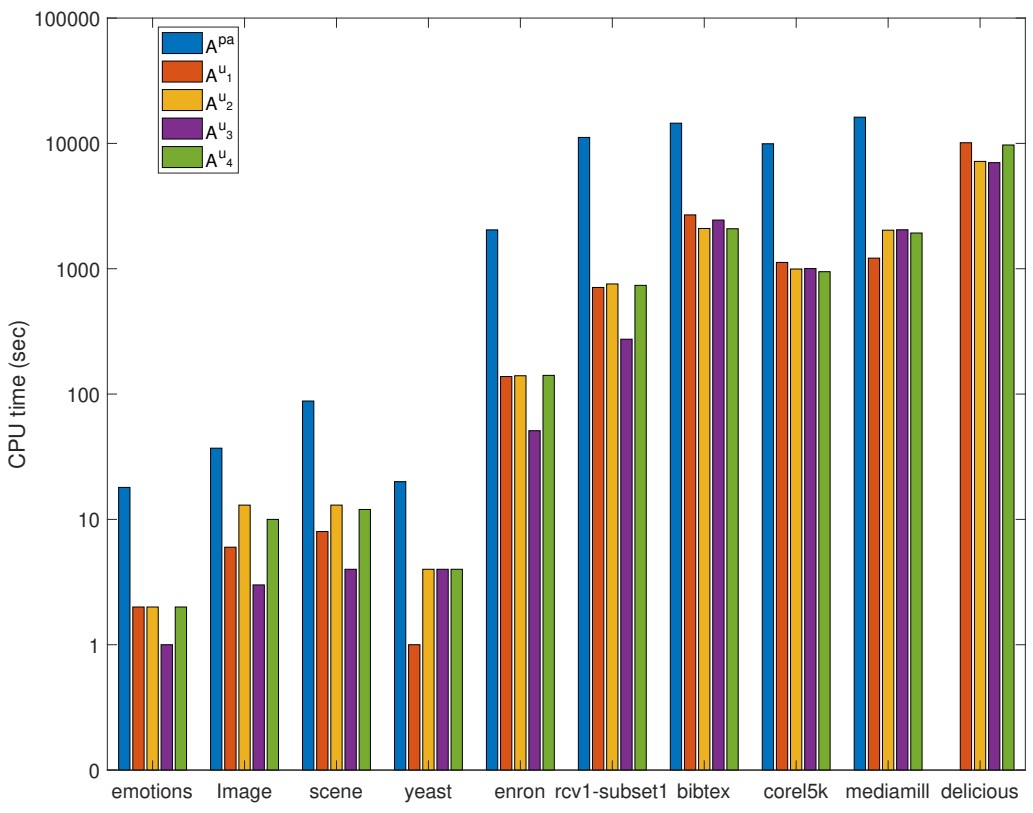

Figure 1: Computational costs of all five algorithms on benchmark datasets.

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

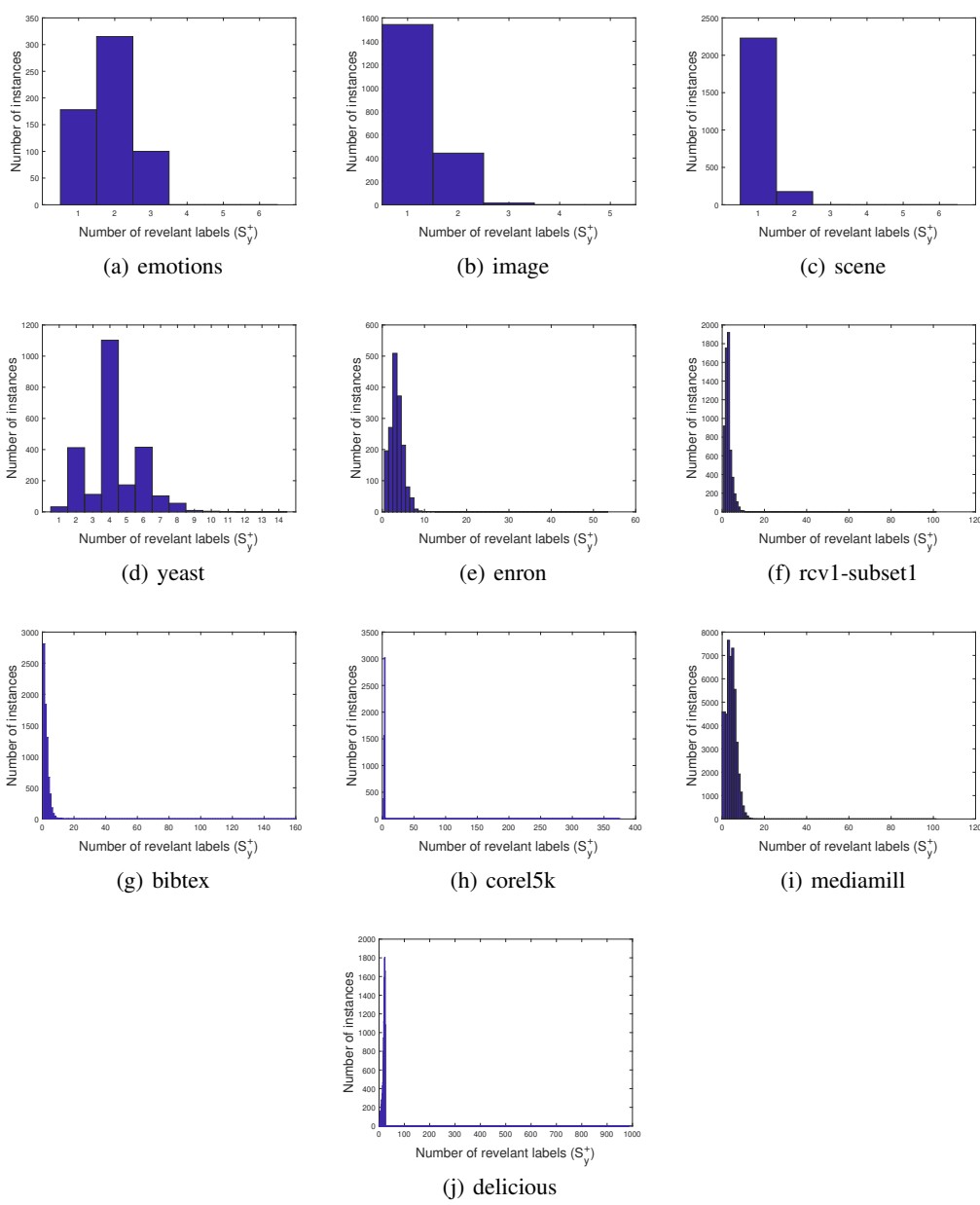

Figure 2: The illustration of the instance-wise class imbalance of each dataset.

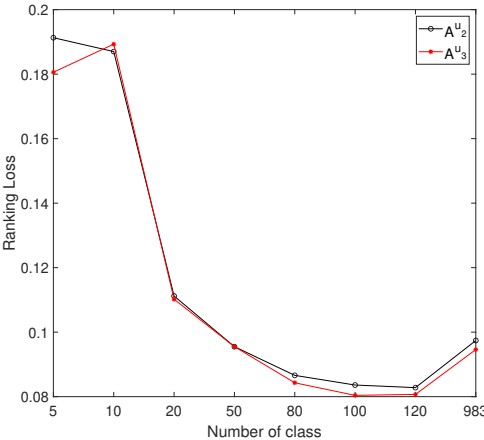

Figure 3: The performance effect w.r.t. the number of class.

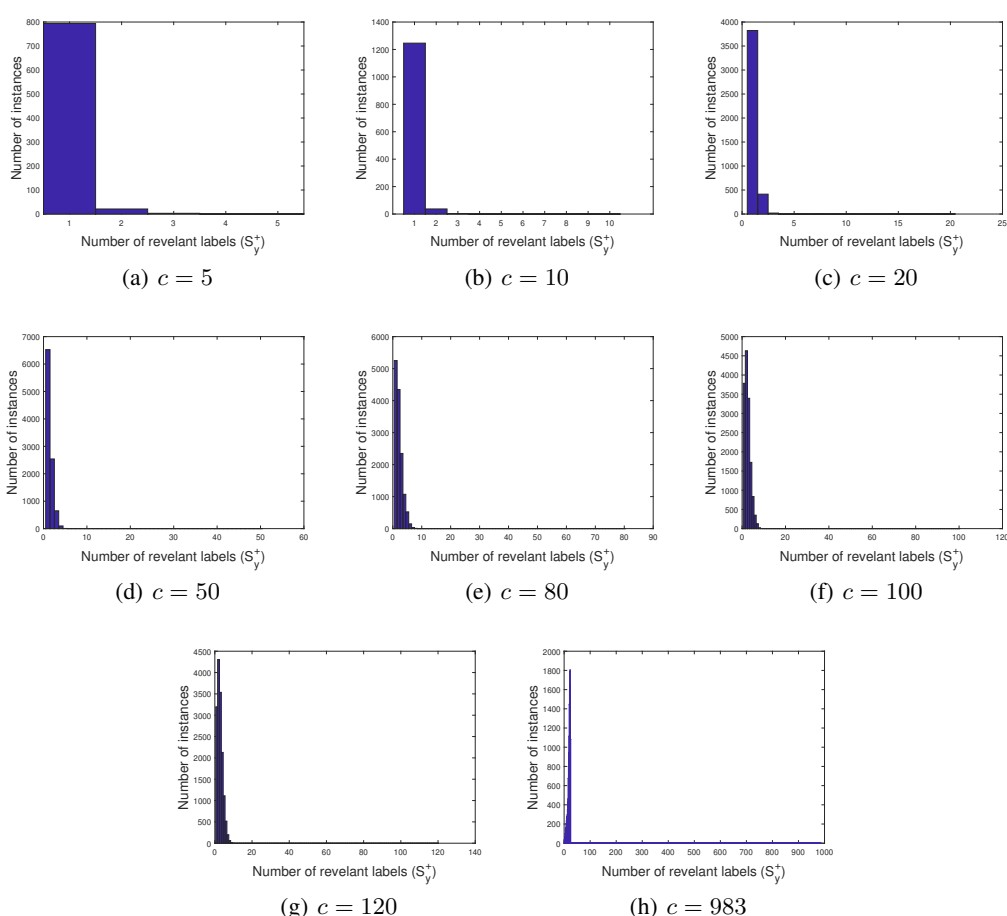

Figure 4: The illustration of the instance-wise class imbalance of the semi-synthetic datasets with randomly selected $c$ labels based on the delicious dataset.

Table 3: The quantitative results about the risks and error bounds on semi-synthetic delicious datasets with randomly selected label size (i.e., 5, 10 ,..., 120). The algorithms are run on the corresponding best hyper-parameters, where $f$ denotes the returned hypothesis of the algorithms. "PUB" denotes the corresponding probabilistic upper bound of the algorithms and we set $\delta = 0.01$ in the error bounds. Note that here we take the validation error as a surrogate of the generalization error, which is reasonable. On each dataset, the best ones w.r.t. $R_{0/1}^r$, $R_{pa}$ and PUB are highlighted in bold.

| | datasets | 5 | 10 | 20 | 50 | 80 | 100 | 120 |
|---|---|---|---|---|---|---|---|---|
| $\mathcal{A}^{pa}$ | $R_{0/1}^r(f)$ | **0.180** | 0.183 | 0.114 | **0.094** | **0.084** | **0.080** | **0.081** |
| | $R_{pa}(f)$ | 0.376 | **0.388** | 0.271 | **0.235** | 0.211 | 0.203 | **0.200** |
| | PUB | **10.80** | **38.09** | **39.59** | **123.6** | **131.1** | 344.9 | 395.1 |
| $\mathcal{A}^{u_2}$ | $R_{0/1}^r(f)$ | 0.193 | 0.186 | **0.111** | 0.097 | 0.087 | 0.083 | 0.083 |
| | $R_{pa}(f)$ | 0.517 | 0.401 | **0.264** | 0.242 | 0.231 | 0.236 | 0.203 |
| | $R_{u_2}(f)$ | 0.487 | 0.271 | 0.128 | 0.065 | 0.042 | 0.034 | 0.023 |
| | PUB | 215.2 | 106.2 | 378.4 | 1341 | 1836 | 2183 | 4685 |
| $\mathcal{A}^{u_3}$ | $R_{0/1}^r(f)$ | 0.181 | **0.182** | 0.112 | 0.096 | **0.084** | **0.080** | **0.081** |
| | $R_{pa}(f)$ | **0.373** | 0.390 | **0.264** | 0.239 | **0.208** | **0.200** | **0.200** |
| | $R_{u_3}(f)$ | 0.871 | 0.991 | 0.787 | 0.798 | 0.731 | 0.713 | 0.712 |
| | PUB | 66.53 | 75.64 | 82.18 | 260.0 | 275.7 | **287.5** | **326.4** |