# OpenReview forum: "Rethinking and Reweighting the Univariate Losses for Multi-Label Ranking: Consistency and Generalization"
_NeurIPS.cc/2021/Conference — NeurIPS 2021 Poster_

### Official Review · Reviewer_z9MC · 2021-07-12

**Rating:** 7
**Confidence:** 5

**Summary:**

This paper attempts to fill the gap between existing theory and practice for Multi-Label Ranking (MLR) problems: why inconsistent pairwise losses often achieve better performance in practice than consistent univariate losses? The authors try to answer this question from the prospective of generalization error bounds, besides the consistency.  Two factors of the distribution (from which the data is drawn) are identified that affect the learning guarantee. Motivated by the observation, the authors propose two new surrogate losses, one of which not only enjoys nearly the similar performance as the popular pairwise surrogate loss, but is also computationally more efficient. The results are well-supported with theoretical results and empirical studies.

Overall I think the paper is clear and well executed.

**Ethical Concerns:**

No ethical issues.

**Limitations And Societal Impact:**

Yes.

**Main Review:**

Thanks for this interesting and well-written paper. Below are comments for each dimension.

Originality.
This paper tries to rigorously study this question in MLR: why inconsistent pairwise losses often achieve better performance in practice than consistent univariate losses? The authors answer the question from the angle of generalization error bounds and consistency analyses. Overall this work is original, though the techniques are standard. Although the relevant papers are cited, the related work section (Section 5) is somewhat inadequate. It would be better if the authors can explain more how this paper is related to the papers discussed in Section 5.

Quality.
The method and the claims are overall correct, though I did not verify the proofs line by line. The empirical study is valid and supports the theoretical claims. The authors are honest about the weakness of their work (e.g. they make it clear that the proposed algorithms are not consistent). My only concern is about comparing upper bounds for the generalization errors (see comment below).

Clarity.
The paper is well-written and well-organized.

Significance.
This paper attempts to solve an interesting, important and practical problem in MLR from the theoretical prospective: why some inconsistent algorithms can often outperform consistent algorithms on real data sets. The question itself is important and valuable to the multi-label community. The study in this paper can potentially inspire new algorithms that not only can perform well in practice, but also enjoy some theoretical guarantees. However, I find that the main technique used (comparing upper bounds for the generalization error) is somewhat limited and may not generalize easily to other settings (e.g. distributions other than extremely imbalanced and balanced).

---

Extra comments:

My concern is how well the upper bound for the generalization error can reflect on the true generalization error. I know that it might be difficult (or even impossible) to directly compare generalization errors of different algorithms, so I can see why the authors choose to compare the upper bounds, and I agree this is a valid approach because a tighter upper bound usually suggests better performance. I have read the improvements the authors made since the ICML 2021 submission, and I can see that the authors have added more discussions about the upper bounds to provide more insights and justifications. I still think it might be better if the authors could show some synthetic experiments justifying the use of upper bounds. The advantage of synthetic experiments is that the generalization error can be calculated, so that one can see if the upper bound for the generalization error can reflect on the true generalization error reasonably well.

---

Typo:

Page 9, line 343: "two extremely cases" -> "two extreme cases"

---

After rebuttal: Thank the authors for providing additional results. I keep my rating for this paper.



**Time Spent Reviewing:**

12 hours

---

> ### Author Response · Authors · 2021-08-09
> **Thank you**
>
> Thank you for the appreciation of our contributions and the valuable comments. We address them in detail below and will further improve in the final version.
>
> ### Q1: Explain more on the related work:
> Thanks for the suggestion. We will add more discussion about the relationship between this paper and the references mentioned in Sec.5.
>
> ### Q2: Generalize the main technique to other settings other than extremely imbalanced and balanced:
> Thanks for your insightful comments. Indeed, it is highly nontrivial to analyze the very general cases. Nevertheless, our analysis can be extended to the case where all data have the same number of relevant labels.
> According to these results, we can see that in an imbalanced case (not necessarily the extremely imbalanced ones), $\mathcal{A}^{pa}$ and $\mathcal{A}^{u_3}$ usually have better upper bounds than $\mathcal{A}^{u_2}$.
>
> Please see our response to reviewer PG73 for details.
>
> ### Q3: Synthetic experiments to justify the use of upper bounds:
> Thanks for your insightful and valuable comments. We agree that it is meaningful to conduct experiments to see whether the upper bound for the generalization error can reflect on the true generalization error reasonably well. We are now trying to calculate the generalization bounds on our semi-synthetic data (in Appendix E) and will compare them to the true generalization errors in the final version.
>
> ### Typos:
> Thanks. We will carefully check this manuscript and fix typos.

---

### Official Review · Reviewer_95pY · 2021-07-13

**Rating:** 7
**Confidence:** 2

**Summary:**

This paper analyzes theoretically the consistency and the generalization error of multilabel ranking algorithms. More specifically, some generalization error bounds are obtained for different loss functions, that are surrogates of the ranking loss or the partial ranking loss.

The analysis is restricted to two extreme cases: (1) instance-wise class balanced distribution (when the number of positive and negative classes is the same for all samples), and (2) extremely imbalanced distribution (when only one class is positive or only one class is negative, for all samples).  It turns out that, while the generalization error bounds for the balanced case are independent on the number of classes, $c$, the  bounds for the imbalanced case are $O(c)$ for some univariate losses (like $L_{u_2}$)  while it is $O(\sqrt{c})$ for pairwise losses (like $L_{pa}$).

Since, in practice, real datasets are highly imbalanced, the theoretical results might explain why $L_{pa}$ shows a better performance than $L_{pa}$ in the experiments with finite datasets, even thought only the latter is consistent.

Based on these findings, the paper proposes some re-weighting versions of the univariate losses with $O(\sqrt{c})$ generalization bounds. The consistency is lost, but some of the reweighting version of $L_{u_2}$ (in particular, $L_{u_3}$) outperform $L_{u_2}$  in the experiments.


**Ethical Concerns:**

There are no ethical issues.

**Limitations And Societal Impact:**

The authors and the reviewer are not aware of potential negative societal impacts.

**Main Review:**

The main contribution of this paper is theoretical. The paper shows some generalization error bounds that can provide a better understanding of multi-label ranking (MLR) algorithms. In particular, the paper highlights the relevance of the class imbalance as a key factor in the behavior of MLR algorithms.

Up to my knowledge, the contribution of the paper is original. Sec. 5 provides some related work. In particular, the reweighting scheme in Eq. (9) was proposed before with base hinge loss, though it is analyzed here in a more general context.

The experimental results are aimed to corroborate the theoretical results, and, in particular, to verify the influence of the re-weighting scheme on the classification performance. Although the value of the experimental work is limited for a practitioner, the paper may suggest that, using a re-weighting scheme for univariate losses is advisable for datasets with large $c$ and class imbalance.

I have checked only about 30% of the proofs, but they are technically correct, under my view.

Summarizing, the paper may have a small expected impact on real applications, but the theoretical contribution is valuable.


**Time Spent Reviewing:**

5

---

> ### Author Response · Authors · 2021-08-09
> **Thank you**
>
> Thank you for the appreciation of our contributions.

---

### Official Review · Reviewer_5DsC · 2021-07-19

**Rating:** 6
**Confidence:** 3

**Summary:**

This paper studies theoretical analysis such as generalization error and consistency of the pairwise and univariate (i.e., pointwise) loss functions for multi-label classification. The main contributions are two folds: (1) propose two reweighted convex surrogate pointwise losses (2) consider the number of labels and the label imbalance into the theoretical analysis.

**Ethical Concerns:**

No ethical concerns.

**Limitations And Societal Impact:**

No negative social impact.

**Main Review:**

In general, the theoretical analysis of generalization error bound and consistency seems technically sound. The writing is clear and the presentation is comfortable to follow. I outline some detailed questions or comments below.

Q1: The assumption of class imbalance in Definition 1 seems too strict. In practice, for large-scale multi-label classification datasets, we often see the number of labels following a long tail distribution and the number of labels per instance is in the order of log(L), where L is the number of labels. Are you able to extend the theoretical analysis for such a case?

Q2: Following Q1, when L is in an extreme scale, such as millions or more, the O(L) computation time complexity for loss functions is prohibitive, and practitioners often resort to various negative sampling methods. While this is not the main focus of this work, perhaps it would be more informative to readers by adding one paragraph regarding this aspect.

Q3: This manuscript would be more strong If there’s some simulated studies that verify the complexity rate of the generalization error as stated in Table 1. For example, we can simulate the imbalance label distribution from some P(x,y), and learn the OVA classifiers with L_{pa}, L_{u_k}, k=1,..,4 and compare their generalization error when the number of class L grows.


**Time Spent Reviewing:**

3

---

> ### Author Response · Authors · 2021-08-09
> **Thank you**
>
> Thank you for the appreciation of our contributions and the valuable comments. We address them in detail below and will further improve in the final version.
>
> ### Q1: Strict assumption of class imbalance in Definition 1 and extension to cases with $\log(L)$ labels per instance:
> Thanks for the inspiring comments. In fact, we can weaken the assumption and extend our analysis to the cases where all instances have the same number of relevant labels including $\log (L)$, while leaving the general case as future work. Please see our response to reviewer PG73 for the details.
>
> ### Q2: Discussion on the extreme scale of $L$:
> Thanks for the suggestion. We agree that it is an important and interesting setting with extremely large $L$, though not the focus of this paper. In fact, our theoretical results indicate that the univariate loss $L_{u_3}$ would probably enjoy better performance than other univariate ones w.r.t. (partial) ranking loss, thus its estimator by use of negative sampling methods may be preferred in practice. We will add the discussion in the final version.
>
> ### Q3: Simulated studies to verify the complexity rate of the generalization error:
> Thanks for your insightful comment.
> To study the effect of the label size $c$, we have conducted experiments on highly imbalanced semi-synthetic datasets with randomly selected $c$ based on the *delicious* dataset (See Fig.3 and Fig.4 in Appendix E for details).
> We found that $\mathcal{A}^{u_3}$ would probably perform better than $\mathcal{A}^{u_2}$ with larger $c$, which confirms our theoretical findings.
>
> However, it is nontrivial to directly plot the complexity rate of $c$ on synthetic data. This is because when we change $c$, we also change the data distribution, while the generalization errors among different data distributions are not directly comparable.
>
> We will add the discussion in the final version.

---

### Official Review · Reviewer_PG73 · 2021-07-20

**Rating:** 7
**Confidence:** 4

**Summary:**

This paper provides theoretical studies on the ranking loss in multi-label learning. For ranking loss, usually, two surrogate losses are used: pairwise and univariate. The paper is motivated by the fact that the pairwise surrogate loss is theoretically unfavored but empirically well performed. To study the surrogate losses, the paper analyzed the original loss function and some proposed reweighting of the loss function by both consistency and generalization analysis. The generalization analysis is conducted by analyzing the upper-bound relationship between different loss functions and the corresponding risks. This analysis shows that the instance-wise class imbalance affects the relationship between these surrogate losses. The paper further analyzed the generalization performance for two extreme cases: extremely imbalanced (all predictions are zero or one for one instance) and absolutely balanced (exact half of the predictions are one for an instance). The paper further analyzed the consistency of these surrogate loss functions and gave some experimental results.

**Limitations And Societal Impact:**

Yes.

**Main Review:**

This paper is a theoretical study, analyzing commonly used surrogate losses for ranking loss in multi-label learning. The paper is motivated by the gap between existing theory and practice for the commonly used pairwise surrogate loss and univariate surrogate loss. I think this direction is novel and worth study. It can contribute to multi-label learning in both theoretical and practical aspects.

The paper proposes two reweighted versions of the univariate losses and analyzed the relationship between these surrogate loss functions, and the generalization error of the algorithms optimizing these loss functions. The most interesting part of the study is it tries to discuss the relationship between the “balance” per instance and the choice of surrogate loss functions. This part provides the most informative insight of the study: pairwise surrogate loss is favoured if data are imbalanced at the instance level. However, the study is based on two extreme cases of imbalance, and there is no study on the in-between cases, or analyze how the surrogate loss’s performance is impacted by a continuous changing imbalance level.

Generally, the paper is clearly written and easy to follow. Especially it gives a thorough intuitive explanation and insights drawn after each formal theoretical result. This may enlarge the impact of the paper to more general readers due to the efficient delivery of its idea.

After rebuttal:
I appreciate the authors' rebuttal in addressing one special case of imbalanced learning out of the two extreme cases in the paper. I keep my positive rate for this paper.


**Time Spent Reviewing:**

4 hours

---

> ### Author Response · Authors · 2021-08-09
> **Thank you**
>
> Thank you for the appreciation of our contributions and the valuable comments. We will further improve in the final version.
>
> As for the in-between cases of imbalance, indeed it is highly nontrivial to consider a continuous changing imbalance level of the distribution when instances may have different numbers of positive labels, and we leave it as an important future direction.
>
> Nevertheless, our framework can be applied to the cases where each instance has the same number of positive labels, denoted as $c_p$. $\frac{\min \\{c_p, \\ c - c_p \\}}{c}$
> directly reflects the imbalance level of the distribution. Note that the extremely imbalanced case ($c_p = 1$ or $c_p = c-1$) and the balanced one ($c_p = c/2$) are included.
>
> Formally, we have the following learning guarantees for the corresponding algorithms:
> $$
> \mathcal{A}^{pa}: R_{0/1}^r (f) \leq	R_{pa} (f) \leq  \hat{R}^{pa}_S(f) + \frac{2 \sqrt{2} \rho}{\sqrt{c_\{min\}}} \sqrt{\frac{c \Lambda^2 r^2}{n}} + 3B \sqrt{\frac{\log \frac{2}{\delta}}{2n}},
> $$
>
> $$
>     \mathcal{A}^{u_2}: R_{0/1}^r (f) \leq R_{pa} (f) \leq (c - c_\{min\}) R_{u_2} (f) \leq (c - c_\{min\}) \hat{R}^{u_2}_S(f) + \frac{2 \sqrt{2} \rho c}{c_\{min\}} \sqrt{\frac{\Lambda^2 r^2}{n}} + \frac{3Bc}{c_\{min\}} \sqrt{\frac{\log \frac{2}{\delta}}{2n}},
> $$
>
> $$
>     \mathcal{A}^{u_3}: R_{0/1}^r (f) \leq R_{pa} (f) \leq R_{u_3} (f) \leq  \hat{R}^{u_3}_S(f) + \frac{4 \rho}{\sqrt{c_\{min\}}} \sqrt{\frac{c \Lambda^2 r^2}{n}} + 6B \sqrt{\frac{\log \frac{2}{\delta}}{2n}},
> $$
>
> where $c_{min} = \min \\{c_p, c - c_p \\}$ for clarity.
>
> According to these results, we can observe that $\mathcal{A}^{pa}$ has an error bound of $O(\sqrt{\frac{c}{c_{min}}})$ that is the same as $\mathcal{A}^{u_3}$, while $\mathcal{A}^{u_2}$ depends on $O(\frac{c}{c_{min}})$. Therefore, in an imbalanced case (not necessarily the extremely imbalanced ones), $\mathcal{A}^{pa}$ and $\mathcal{A}^{u_3}$ usually have better bounds than $\mathcal{A}^{u_2}$.
>
> We will add these results and discussion in the final version.

---

### Decision · Program_Chairs · 2021-09-27

**Decision:**

Accept (Poster)

**Comment:**

The reviewers and I agree that this paper offers several valuable insights, both theoretically and empirically, into a class of methods for multi-label classification. I also agree with a reviewer that error bounds are not necessarily the best proxies for the actual error, and that the authors should include additional content on this point, e.g., through simulation studies as suggested by the reviewer, in the final version.